# Syntaxin-6 delays prion protein fibril formation and prolongs the presence of toxic aggregation intermediates

**Daljit Sangar, Elizabeth Hill, Kezia Jack, Mark Batchelor, Beenaben Mistry, Juan M Ribes, Graham S Jackson, Simon Mead, Jan Bieschke***

MRC Prion Unit at UCL, Institute of Prion Diseases, London, United Kingdom

**Abstract** Prions replicate via the autocatalytic conversion of cellular prion protein (PrP^C) into fibrillar assemblies of misfolded PrP. While this process has been extensively studied in vivo and in vitro, non-physiological reaction conditions of fibril formation in vitro have precluded the identification and mechanistic analysis of cellular proteins, which may alter PrP self-assembly and prion replication. Here, we have developed a fibril formation assay for recombinant murine and human PrP (23-231) under near-native conditions (NAA) to study the effect of cellular proteins, which may be risk factors or potential therapeutic targets in prion disease. Genetic screening suggests that variants that increase syntaxin-6 expression in the brain (gene: STX6) are risk factors for sporadic Creutzfeldt–Jakob disease. Analysis of the protein in NAA revealed, counterintuitively, that syntaxin-6 is a potent inhibitor of PrP fibril formation. It significantly delayed the lag phase of fibril formation at highly sub-stoichiometric molar ratios. However, when assessing toxicity of different aggregation time points to primary neurons, syntaxin-6 prolonged the presence of neurotoxic PrP species. Electron microscopy and super-resolution fluorescence microscopy revealed that, instead of highly ordered fibrils, in the presence of syntaxin-6 PrP formed less-ordered aggregates containing syntaxin-6. These data strongly suggest that the protein can directly alter the initial phase of PrP self-assembly and, uniquely, can act as an 'anti-chaperone', which promotes toxic aggregation intermediates by inhibiting fibril formation.

*For correspondence:
j.bieschke@ucl.ac.uk

**Competing interest:** The authors declare that no competing interests exist.

## Editor's evaluation

The current study presents an important discovery about Syntaxin 6 (Stx6)'s anti-chaperoning activity on PrP. The authors provide compelling evidence that the anti-chaperone activity arises as Stx6 delays PrP fibril formation and in the presence of Syntaxin 6, the amorphous aggregates of PrP are more toxic to neuronal cells. This study provides a critical molecular link between PrP aggregation and neurotoxicity.

## Introduction

Prion replication and prion-like mechanisms are believed to drive dozens of human neurodegenerative and systemic diseases, as well as scrapie, bovine spongiform encephalopathy, and chronic wasting disease in animals (*Prusiner, 1998*; *Brundin et al., 2010*; *Frost and Diamond, 2010*). All of these disorders are caused by protein misfolding, but the details of the self-replication mechanism, the cellular factors involved in prion replication, and the molecular basis for neurotoxicity are far from clear despite decades of research (*Collinge and Clarke, 2007*). Prions are fibrillar assemblies of misfolded cellular prion protein (PrP^C), which form highly ordered parallel in register intermolecular β-sheet amyloid structures (prion rods, *Manka et al., 2022*; *Kraus et al., 2021*). While purified

prion rods are not directly neurotoxic, prion-associated toxicity can be blocked by anti-PrP antibodies (*Benilova et al., 2020*; *Reilly et al., 2022*) suggesting that non-fibrillar PrP assemblies, which form after prion titers plateau (*Sandberg et al., 2011*; *Sandberg et al., 2014*; *Collinge, 2016*), could be the toxic agents, as is the case for other amyloid-forming proteins (*Haass and Selkoe, 2007*; *Corbett et al., 2020*).

Prions can be highly infectious. However, accidental exposure only accounts for a tiny percentage of Creutzfeldt–Jakob disease (CJD) cases. Similarly, mutations in the *PRNP* gene, which codes for the prion protein PrP, cause only 10–15% of CJD cases (*Mead et al., 2019*). The majority of prion disease cases are sporadic in nature, where PrP is thought to spontaneously misfold. It is likely that hidden genetic risk factors modulate either the susceptibility to prion replication, prion-associated neurotoxicity, or the initial prion formation. A recent genome-wide association study to discover non-*PRNP* risk factors of sporadic CJD (sCJD) identified two highly significant risk loci including one in and near to *STX6* (*Jones et al., 2020*). Elevated *STX6* mRNA in the caudate and putamen nuclei, two regions implicated in CJD pathology (*Meissner et al., 2009*), correlated with CJD risk.

*STX6* encodes syntaxin-6, an 8-exon, 255 amino-acid protein that predominantly localizes to the trans-Golgi network (TGN), and is mainly involved in recycling of cargo between the TGN and early endosomes (*Jung et al., 2012*). Syntaxin-6 is thought to form part of the t-SNARE complex involved in the decision of a target membrane to accept the fusion of a vesicle (*Wendler and Tooze, 2001*). Misfolded prion protein in infected cells localizes at the plasma membrane, the likely site of prion conversion (*Goold et al., 2011*), as well as in perinuclear compartments. Early and late endosomes, the endocytic recycling compartment and multivesicular bodies have also been implicated as prion replication sites (*Yamasaki et al., 2018*; *Yim et al., 2015*). These observations raise two alternative mechanistic hypotheses for the role of syntaxin-6 in prion disease: (1) a direct interaction with PrP in the course of misfolding and prion replication, or (2) an indirect effect through the cellular processing of PrP$^C$ or prions.

*STX6* is also a risk gene for progressive supranuclear palsy (*Höglinger et al., 2011*), where the protein interacts directly with tau through its transmembrane domain (*Lee et al., 2021*). STX6 has been identified in a proteome-wide association study in Alzheimer's disease (*Wingo et al., 2021*). Downregulation of syntaxin-6 expression in a murine neuroblastoma scrapie cell model (PK1) did not alter PrP$^C$ expression and had no obvious effect on PrP$^C$ trafficking (*Jones et al., 2020*), suggesting that it may directly alter prion replication by a yet undiscovered chaperone-like activity.

An array of chaperone proteins aid and tightly control protein folding and protein homeostasis in the cell (*Hartl, 2011*; *Balch et al., 2008*). Chaperones, such as the heat shock protein HSP70, inhibit protein misfolding and can delay amyloid formation in vitro (*Glover and Lindquist, 1998*) and in model organisms (*Thackray et al., 2022*). It seems likely that more proteins with chaperone-like functions are yet to be discovered.

Assessing the role of proteins with chaperone-like activity in PrP misfolding, fibril formation, and toxicity requires assays under native or near-native conditions. The established fibril formation assays for PrP in vitro require non-physiological conditions, such as high temperatures (*Atarashi et al., 2011*), denaturants (*Legname et al., 2004*), and detergents or low pH (*Post et al., 1998*). These conditions facilitate the unfolding and conversion of PrP$^C$, but preclude the analysis of other folded proteins in the context of PrP fibril formation. Here, we have developed a fibril formation assay for recombinant murine and human PrP$^C$ (23-231) under near-native conditions to study the effect of syntaxin-6 and other cellular factors, which may be risk factors or potential therapeutic targets in prion disease.

## Results

### Full-length prion protein forms amyloid fibrils under near-native conditions

In order to study PrP fibril formation under near-native conditions in vitro, full-length murine PrP$^C$ 23-231 (mPrP23) and human PrP$^C$ 23-231 (hPrP23) were expressed recombinantly (*Figure 1—figure supplement 1A*), purified via Ni-NTA chromatography, and refolded into their native conformation (*Figure 1—figure supplement 1E and F*). Proteins were incubated in a non-binding 96-well plate under agitation, and amyloid formation was monitored via thioflavin T (ThT) fluorescence (*Figure 1A*, *Figure 1—figure supplement 2*). We tested different temperatures pH values and agitation conditions

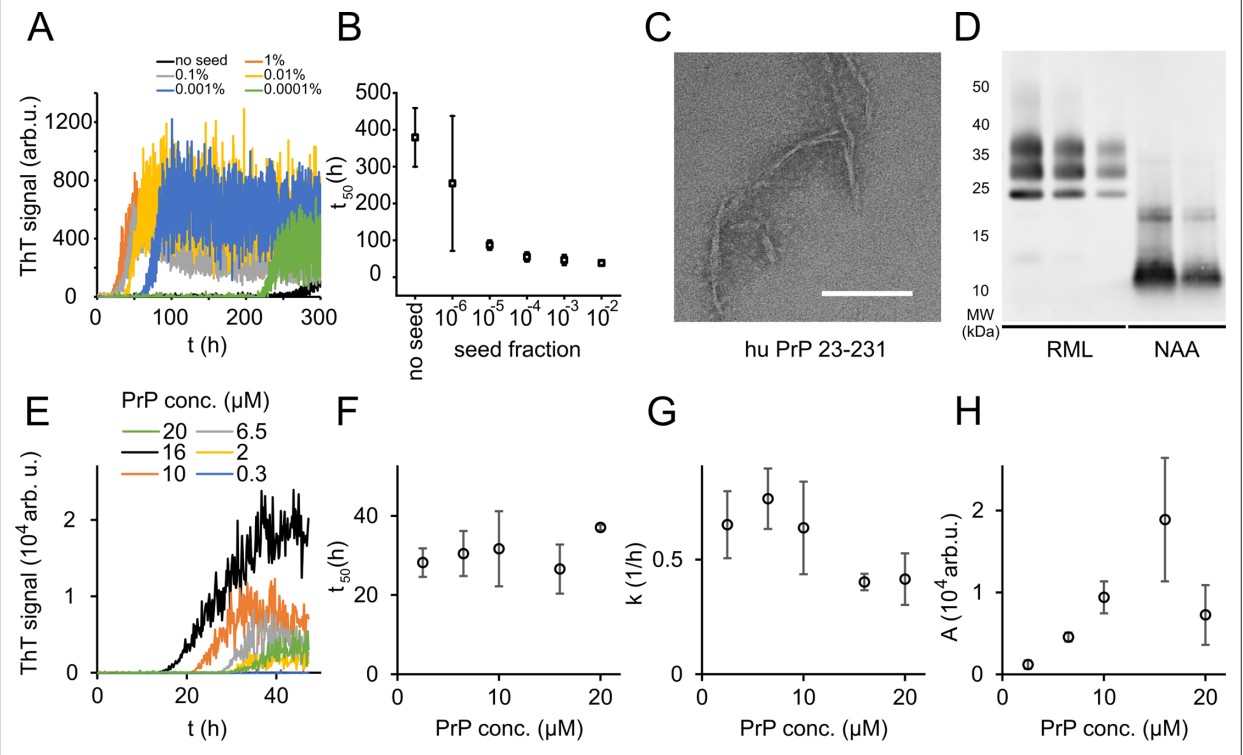

**Figure 1.** Native aggregation assay (NAA) of human PrP (23-231). (**A**) Seed titration experiment using seed generated from hPrP23 fibrils formed de novo in NAA, seed fraction (w/w), 2.5 µM hPrP23, pH 6.8, 42°C. (**B**) Plot of lag phase vs. fraction of seed, mean ± SD, n = 3. (**C**) TEM image of hPrP23 fibrils formed after 166 hr incubation. (**D**) Western blot (ICSM35) of ex vivo RML prions and NAA aggregation endpoint samples after digestion with proteinase K (50 µg/ml, 30 min) and NaPTA precipitation. (**E–H**) Titration of PrP concentration (0.3–20 µM) in ThT aggregation assays containing 0.01% seed, pH 6.8, 42°C. (**E**) Plot of ThT fluorescence vs. time. (**F**) Plot of lag time $t_{50}$ vs. PrP concentration. (**G**) Plot of elongation rate constant $k$ vs. PrP concentration. (**H**) Plot of fluorescence amplitude $A$ vs. PrP concentration. Source data is available at https://doi.org/10.17632/yggpkrgnx8.1.

The online version of this article includes the following figure supplement(s) for figure 1:

**Figure supplement 1.** Protein characterization.

**Figure supplement 2.** Optimization of native aggregation assay (NAA).

**Figure supplement 3.** mPrP23 seeding kinetics.

**Figure supplement 4.** Amylofit analysis of concentration-dependent native aggregation assay (NAA) data.

to assess whether PrP could form amyloid fibrils in vitro under near-native conditions (***Figure 1—figure supplement 2A–C***). While PrP was slow to form aggregates de novo without the addition of seeds, elevated temperatures accelerated fibril formation (***Figure 1—figure supplement 2A***) as had been previously observed (***Ziaunys et al., 2021***). Aggregation at 42°C, pH 6.8 yielded highly reproducible conditions for the formation of fibrils (***Figure 1A–C***, ***Figure 1—figure supplement 2B and C***) when agitated by glass or silanized zirconia beads. Fibrils had diameters of 8–12 nm and morphologies typical of amyloid fibrils (***Figure 1C***, ***Figure 1—figure supplement 2G***). Fibrils formed in native aggregation assay (NAA) were resistant to digestion by proteinase K to the same degree as authentic prion rods prepared from RML-infected mice, producing a main PK-resistant fragment of ~12 kD (***Figure 1D***, ***Figure 1—figure supplement 2H***). The addition of preformed fibrillar aggregates as seeds accelerated fibril formation in a concentration-dependent manner by decreasing lag-times ($t_{50}$, ***Figure 1A and B***, ***Figure 1—figure supplements 2D and 3***) but had no significant influence on rates of fibril elongation (***Figure 1—figure supplement 2E***). Lag phases and elongation rates of mPrP23 in vitro were indistinguishable when using synthetic seed formed from recombinant PrP^C or using authentic prion rods, isolated from mice infected with scrapie strains RML or ME7 in (***Figure 1—figure supplements 2D, E and 3***). Interestingly, fibrils formed from synthetic seeds displayed increased ThT fluorescence when compared to RML and ME7 seeded fibrils, hinting at a difference in fibril structure

imprinted by the different seed templates (*Figure 1H*, *Figure 1—figure supplement 2F*). For safety reasons, seeding of hPrP23 was restricted to synthetic PrP fibrils.

The lag phase of hPrP23 aggregation was largely independent of PrP monomer concentration (*Figure 1E and F*), while elongation rates even showed a slight inverse reaction to monomer concentration (*Figure 1G*). Therefore, PrP refolding rather than elongation may be rate-limiting under the assay conditions. ThT amplitude at the end of growth was proportional to monomer concentration (*Figure 1H*). However, the lag phase increased and ThT amplitude dropped at monomer concentrations above 16 μM, suggesting the formation of off-pathway PrP aggregates at high monomer concentrations, which do not contribute to fibril formation (*Powers and Powers, 2008*).

Analysis of aggregation kinetics using the Amylofit framework confirmed this interpretation (*Figure 1—figure supplement 4*). The concentration dependence of fibril formation could be described adequately only under the assumption that kinetics were dominated by saturated elongation. It should be noted that kinetic curves in the NAA have higher fluctuations in the ThT signal than found under denaturing conditions (see *Sun et al., 2023*). This is most likely due to the formation of large PrP aggregates in the NAA. These fluctuations had no influence on the accuracy of kinetic fitting when comparing lag times derived from raw data (*Figure 1E and F*) with those from smoothed aggregation curves (*Figure 1—figure supplement 4A and B*).

## Syntaxin-6 inhibits PrP fibril formation

The *STX6* gene is a proposed risk factor for sCJD, suggesting that it may alter PrP fibril formation. We therefore tested what effect the syntaxin-6 protein had on PrP fibril formation under native conditions and compared it to two reference proteins: a well-characterized amyloid inhibitor, the heat shock protein HSPA1A, which is the human, inducible form of HSP70, and the α-helical microtubule-associated protein stathmin 1 (STMN1) as positive and negative controls, respectively (*Figure 1—figure supplement 1A*). Circular dichroism confirmed that all proteins retained their native, mostly α-helical folds under assay conditions (*Figure 1—figure supplement 1B–F*).

To our surprise, syntaxin-6 delayed the lag phase of hPrP23 fibril formation even at highly substoichiometric molar ratios of 1:100 and prevented fibril formation entirely at equimolar ratio (STX6, *Figure 2A and B*). At the same time, it only slightly lowered fibril elongation rates (*Figure 2C*). Syntaxin-6 promoted the formation of aggregate clusters (*Figure 2D and E*) rather than distinct amyloid fibrils (*Figure 1C*). Immuno-gold staining revealed that syntaxin-6 was incorporated into these aggregates (*Figure 2E*). Similarly, HSPA1A prevented fibril formation in favor of amorphous PrP aggregates (*Figure 2—figure supplement 1A*), while hPrP23 formed fibrillar aggregates in the presence of control protein STMN1 (*Figure 2—figure supplement 1B*). However, sedimentation after 0–90 hr incubation revealed that syntaxin-6 delayed but did not prevent the formation of hPrP23 aggregates. In contrast, a substantial fraction of hPrP23 remained soluble when incubated with HSPA1A at a 1:10 molar ratio (*Figure 2—figure supplement 2*).

To probe the nature of PrP aggregates formed in the presence of syntaxin-6, we performed two secondary seeding assays. In the first assay, we tested the seeding capacity of PrP samples harvested at different incubation times (20, 40, 60, 90 hr) in the presence and absence of syntaxin-6 in a NAA seeded with 0.1% fibrillar hPrP23 (*Figure 2—figure supplement 3*). Samples harvested during the lag phase of either reactions were not seeding competent, but both aggregates formed in the presence and absence of syntaxin-6 (1:10) were seeding competent after the formation of ThT-positive aggregates, indicating that, at sub-stoichiometric ratios, syntaxin-6 delays, but does not prevent the formation of seeding competent PrP fibrils. We then generated seeds in a second experiment, in which the primary seed was added at a lower concentration (0.01%), which prolonged the delay in fibril formation by syntaxin-6 (*Figure 2—figure supplement 4A*). Seed preparations harvested after 70 hr incubation were then separated into total, soluble, and insoluble fractions and added to secondary seeding assays at $10^{-3}$ to $10^{-8}$ molar ratio monomer equivalents (*Figure 2—figure supplement 4B–E*). Both in syntaxin-6 and in untreated samples, seeding activity was only found in the total and insoluble fractions. The presence of syntaxin-6 lowered the amount of seeding competent aggregates by at least three orders of magnitude (*Figure 2—figure supplement 4C–E and 5*).

To further probe the mechanism of syntaxin-6, we tested the influence of NaCl concentration on fibril formation in the presence and absence of syntaxin-6 (1:10 molar ratio) in the NAA (*Figure 2—figure supplement 6*, *Figure 3—figure supplements 2 and 3*). We compared kinetics in the presence

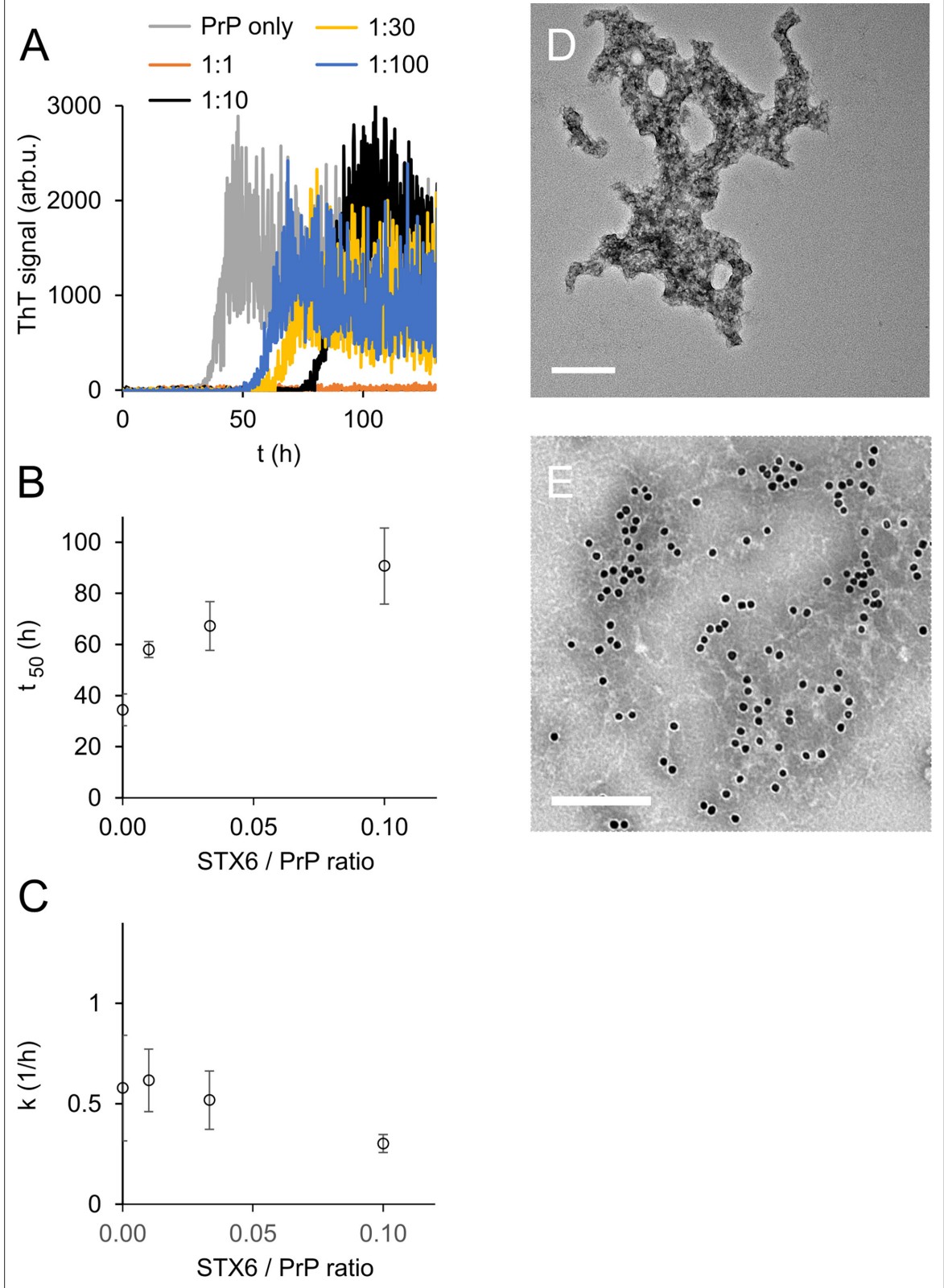

**Figure 2.** Syntaxin-6 (STX6) delays hPrP23 fibril formation at molar ratios of 1:1–1:100; 2.5 µM hPrP23, pH 6.8, 42°C, 0.01% seed. (**A**) ThT fluorescence vs. time. (**B**) Plot of lag phase $t_{50}$ vs. molar ratio of STX6/PrP; mean ± SD, n = 3. (**C**) Plot of elongation rate constant $k$ vs. molar ratio of syntaxin-6/PrP. (**D**) TEM image of hPrP23 co-aggregated with syntaxin-6 at 1:10 (STX6/PrP) molar ratio under standard native aggregation assay (NAA) conditions for

*Figure 2 continued on next page*

*Figure 2 continued*

116 hr. (**E**) Immuno-TEM image of hPrP23 – syntaxin-6 co-aggregate cluster after 100 hr aggregation. Syntaxin-6 is labeled with anti-syntaxin-6 Ab/10 nm anti-rabbit immunogold beads; scale bars 200 nm.

The online version of this article includes the following figure supplement(s) for figure 2:

**Figure supplement 1.** TEM images of native aggregation assay (NAA) aggregation endpoints of hPrP23 co-aggregated with HSP70 (HSPA1A) or stathmin 1 (STMN1).

**Figure supplement 2.** Sedimentation assay of hPrP23.

**Figure supplement 3.** Seeding capacity of aggregation intermediates.

**Figure supplement 4.** Secondary seeding seed dilution experiment of hPrP23 aggregates.

**Figure supplement 5.** Secondary seeding seed dilution kinetics.

**Figure supplement 6.** Salt-dependent inhibition of fibril formation.

of syntaxin-6 to those in the presence of HSPA1A and STMN1, respectively, at the same molar ratio (*Figure 2—figure supplement 6A*). HSPA1A delayed fibril formation to a similar degree as syntaxin-6, whilst, as expected, STMN1 had no significant effect on fibril formation kinetics. In the absence of syntaxin-6, the lag phase of PrP fibril formation decreased only weakly with NaCl concentration while the elongation rate increased (*Figure 2—figure supplement 6B and C*). Syntaxin-6 prolonged the lag phase at low and physiological salt concentrations, but lag phases decreased with increasing NaCl concentration, suggesting that ionic interactions may mediate the binding of syntaxin-6 to PrP. A similar salt dependence was observed for HSPA1A (*Figure 2—figure supplement 6D*), while STMN1 did not inhibit fibril formation at any salt concentration (*Figure 2—figure supplement 6E*). Syntaxin-6 did not significantly reduce fibril elongation compared with untreated PrP independently of salt concentration (*Figure 2—figure supplement 6C*).

To visualize the interaction of syntaxin-6 with PrP aggregates and fibrils, we imaged NAA endpoints by dSTORM super-resolution microscopy (*Figure 3—figure supplement 1*) revealing that hPrP23 and syntaxin-6 co-aggregated into large aggregate clusters. Super-resolution microscopy indicated a fibrillar substructure of these aggregates (*Figure 3—figure supplement 1A and B*) similar to that observed in TEM (*Figure 2D and E*). When, conversely, syntaxin-6 labeled with AlexaFluor647 was added to pre-formed fibrillar hPrP23 formed under NAA conditions, it preferentially bound to specific 'hotspots' on the PrP fibril (*Figure 3A and B*, indicated by arrows). In this experiment, hPrP23 fibrils were unlabeled and were visualized by transient amyloid binding (TAB) imaging using the amyloidophilic dye Nile red (*Spehar et al., 2018*). We repeated the experiment with syntaxin-6-AlexaFluor488 (*Figure 3E*) and with unlabeled proteins using immuno-TEM (*Figure 3F*) to exclude that the interaction was influenced by the choice of fluorescent dye. Both assays confirmed the presence of interaction hotspots. Interestingly, interaction sites were often located at fibril ends or kinks in the fibril. Since syntaxin-6 only weakly affected apparent fibril elongation rates in bulk assays, this may suggest a role of syntaxin-6 in fibril breakage or secondary nucleation. PrP formed longer, unbranched fibrils with increasing salt concentration. As had been seen in TEM, syntaxin-6 induced the formation of aggregate clusters when visualized by TAB imaging (*Figure 3—figure supplements 2 and 3*) when compared with untreated fibrils, supporting the hypothesis that the protein may alter secondary nucleation/branching of fibrils.

## Syntaxin-6 interacts with PrP in cell models of prion disease

Syntaxin-6 and PrP need to interact in order to directly affect prion replication and/or prion-associated toxicity in vivo. We therefore probed the interaction of syntaxin-6 with PrP via Förster resonance energy transfer (FRET) imaging in the PK1 cell model (*Klöhn et al., 2003*), which can be persistently infected with RML prions. Proteins were stained with anti-syntaxin-6 antibody (C34B2)/anti-rabbit-Alx647 and three monoclonal anti-PrP antibodies (5B2, 6D11, 8H4)/anti-mouse-RhX, which bound different regions of PrP (*Figure 4A*, *Figure 4—figure supplements 1 and 2*). PixFRET analysis indicated a perinuclear interaction between both proteins, both in non-infected (PK1) and infected (iS7) cells (*Figure 4A*, wide arrows). An additional FRET signal on the plasma membrane of infected cells suggests that syntaxin-6 may be recruited into misfolded PrP assemblies (*Figure 4A*, narrow arrows). This interaction resulted in FRET signals with all three anti-PrP antibodies, whereas only the two antibodies directed to the unfolded N-terminal domain of PrP (5B2 and 6D11) displayed FRET in

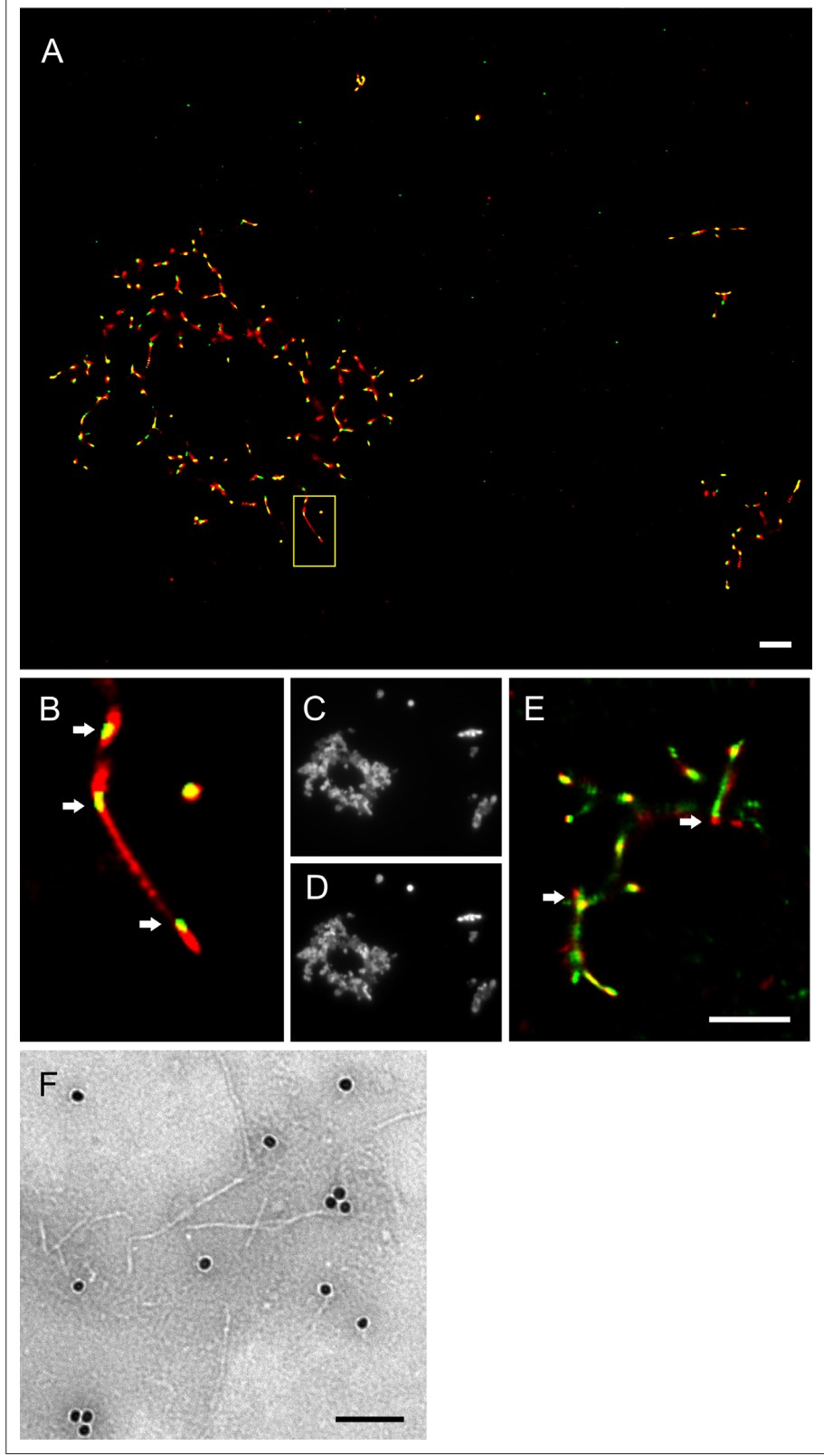

**Figure 3.** Imaging of hPrP23 fibrils incubated with syntaxin-6. hPrP23 (2.5 µM) was pre-aggregated in native aggregation assay (NAA) for 115 hr and incubated with syntaxin-6 (250 nM) for 1 hr. hPrP was visualized by TAB imaging using 10 nM Nile red dye; syntaxin-6 was labeled with AlexaFluor488 (**A–D**) or AlexaFluor647 (**E**) and imaged by dSTORM. (**A**) SR image overlay shows syntaxin-6 binding at hotspots and PrP fibril ends. (**B**) Magnified

*Figure 3 continued on next page*

*Figure 3 continued*

area from (**A**). (**C**) Widefield image taken with green laser (561 nm) illumination. (**D**) Widefield image taken under blue laser (473 nm) illumination. (**E**) SR image overlay of PrP and syntaxin-6-AF647 images; scale bars 2 µm. (**F**) TEM Immuno-gold staining of hPrP23 fibrils incubated with syntaxin-6; scale bar 100 nm.

The online version of this article includes the following figure supplement(s) for figure 3:

**Figure supplement 1.** Overlay of dSTORM SR images taken at 473 nm and 638 nm excitation.

**Figure supplement 2.** TAB SR microscopy images of aggregation endpoints of hPrP23.

**Figure supplement 3.** TAB SR microscopy images of aggregation endpoints of hPrP23 co-aggregated with syntaxin-6.

---

membrane-associated compartments (*Figure 4—figure supplement 1*). This may suggest a different binding mode of syntaxin-6 to PrP species in both compartments.

## Syntaxin-6 knockout does not affect the replication of prions in vitro

Infectious prions can be replicated in cell-free conditions using protein misfolding cyclic amplification (PMCA), whereby the conversion of PrP$^C$ into PK-resistant PrP$^{Sc}$ is enhanced by cyclic bursts of sonication in a brain homogenate substrate (*Bieschke et al., 2004*). Here, we employed PMCA to explore whether the presence or absence of syntaxin-6 in the substrate affected prion conversion. To this end, we seeded PMCA substrates derived from *Stx6*$^{+/+}$ and *Stx6*$^{-/-}$ mice, respectively, (*Hill, 2023*) with RML prions. Both substrates generated comparable amounts of PK-resistant PrP (*Figure 4B*), indicating that syntaxin-6 does not directly alter prion replication in vitro under PMCA conditions, which are dominated by fibril fragmentation/elongation.

## Syntaxin-6 prolongs the presence of toxic PrP aggregation intermediates

The previous data would suggest a protective rather than deleterious effect of syntaxin-6 in prion disease. However, the oligomer toxicity hypothesis posits that mature fibrils may not be the toxic species in prion nor in other amyloid diseases (*Collinge, 2016*; *Haass and Selkoe, 2007*). Correspondingly, purified, highly infectious prion rods were not toxic to primary neurons (*Benilova et al., 2020*). To test whether the same logic applied to the effect of syntaxin-6 on PrP aggregation, we assessed the toxicity of PrP to mouse primary neurons at different stages of the aggregation kinetics by neurite length (*Figure 5*) and by counting the number of viable neurons (*Figure 5—figure supplement 1*). PrP aggregates were highly toxic to primary neurons during lag and early growth phases (20 hr and 40 hr incubation), but neurotoxicity was diminished at the plateau phase of fibril formation, so that endpoint aggregates (90 hr) were no more toxic than PrP monomers or buffer controls (*Figure 5A and C*, *Figure 5—figure supplement 1*). Notably, PrP toxicity preceded the formation of seeding-competent assemblies (*Figure 2—figure supplement 3*), supporting the hypothesis that the toxic PrP species is a pre-fibrillar assembly.

In contrast, PrP aggregated in the presence of syntaxin-6 (1:0.1 molar ratio) retained its neurotoxicity significantly longer than PrP incubated in the absence of syntaxin-6 (*Figure 5B and C*, *Figure 5—figure supplement 1*). While PrP species formed early in the lag phase of aggregation at 20 hr were neurotoxic both in the presence and absence of syntaxin-6 when compared to t = 0 hr, PrP was significantly more toxic in the presence of syntaxin-6 at later time points when PrP would form fibrils in the absence of syntaxin-6 (*Figure 5C*, *Figure 5—figure supplement 1*). Thus, in delaying or preventing PrP fibril formation, syntaxin-6 prolonged the presence of toxic aggregation intermediates and exacerbated neurotoxicity.

## Discussion

### Development of a novel native aggregation assay

We developed a fibril formation assay for full-length mouse and human PrP$^C$ to be able to study the mechanistic effect of cellular proteins such as the proposed sCJD risk factor syntaxin-6 on PrP misfolding and self-assembly under near-native conditions. A multitude of in vitro fibril formation assays have been developed previously (*Schmidt et al., 2015*). While the generation of authentic

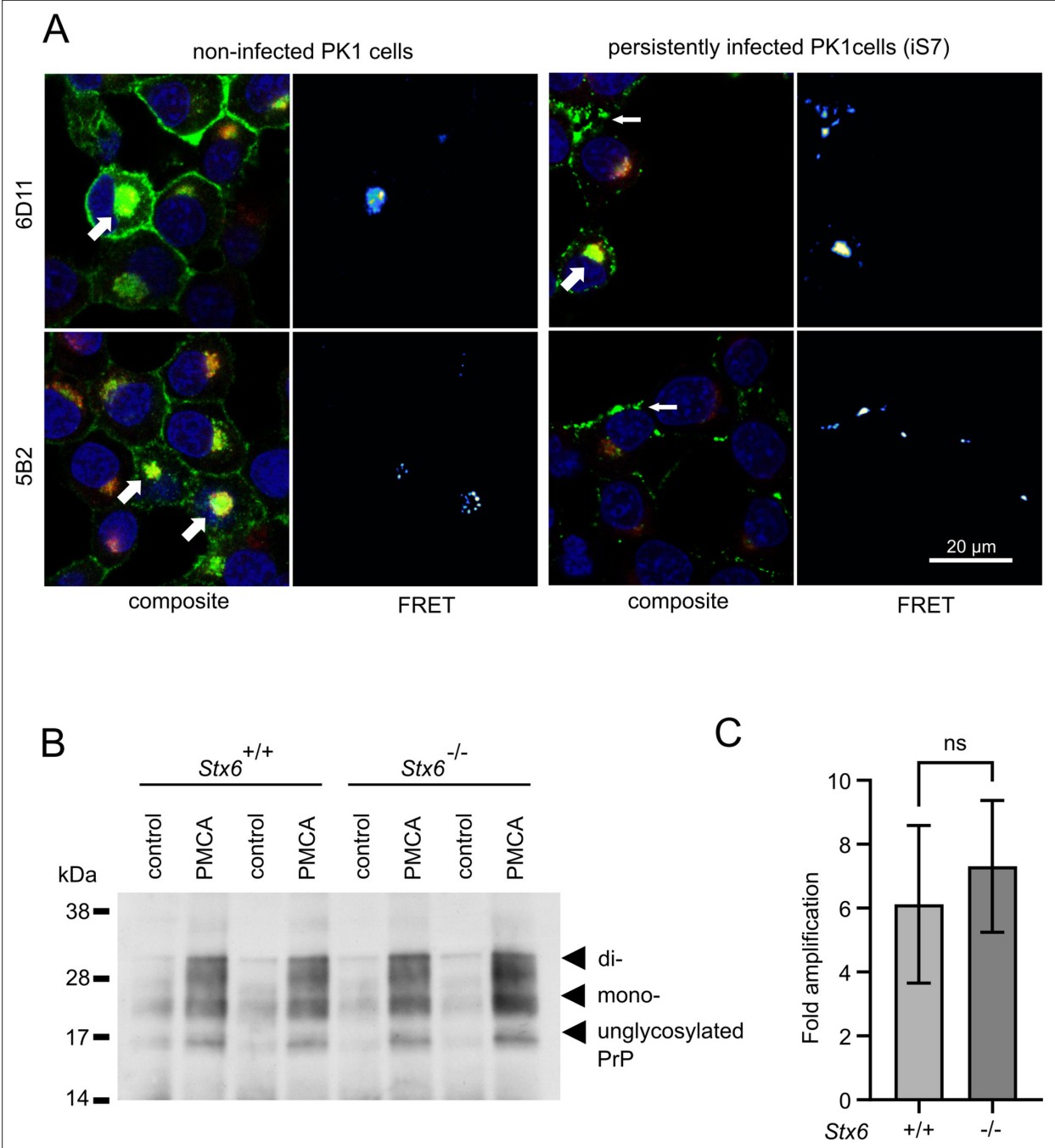

**Figure 4.** Interaction of PrP and syntaxin-6 in vivo. (**A**) Non-infected PK1 cells and persistently infected PK1 cells (iS7) were immuno-stained with anti-PrP antibodies 6D11 and 5B2 (green), with anti-syntaxin-6 antibody (red) and with DAPI (blue). Förster resonance energy transfer (FRET) analysis reveals interaction in perinuclear compartments (wide arrows) and at membranes in infected cells (narrow arrows). Panels show zoomed regions of images in *Figure 4—figure supplement 1*. (**B, C**) In vitro prion replication by protein misfolding cyclic amplification (PMCA) using $Stx6^{+/+}$ and $Stx6^{-/-}$ mouse brains as substrate. PMCA reactions were seeded with RML prions from terminally ill mice and subjected to PMCA for 96 cycles over 48 hr. (**B**) Representative western blot (ICSM35) after PK digestion. Molecular weight markers are indicated on the left. (**C**) The PrP^Sc signal was quantified using densitometry and normalized to the control unamplified reaction. Bar graphs each represent mean ± SD of biological replicates from three separate mice, each blotted as two technical replicates. Source data is available at https://doi.org/10.17632/yggpkrgnx8.1.

The online version of this article includes the following figure supplement(s) for figure 4:

**Figure supplement 1.** hPrP/syntaxin-6 Förster resonance energy transfer (FRET) analysis.

**Figure supplement 2.** hPrP/syntaxin-6 Förster resonance energy transfer (FRET) analysis controls.

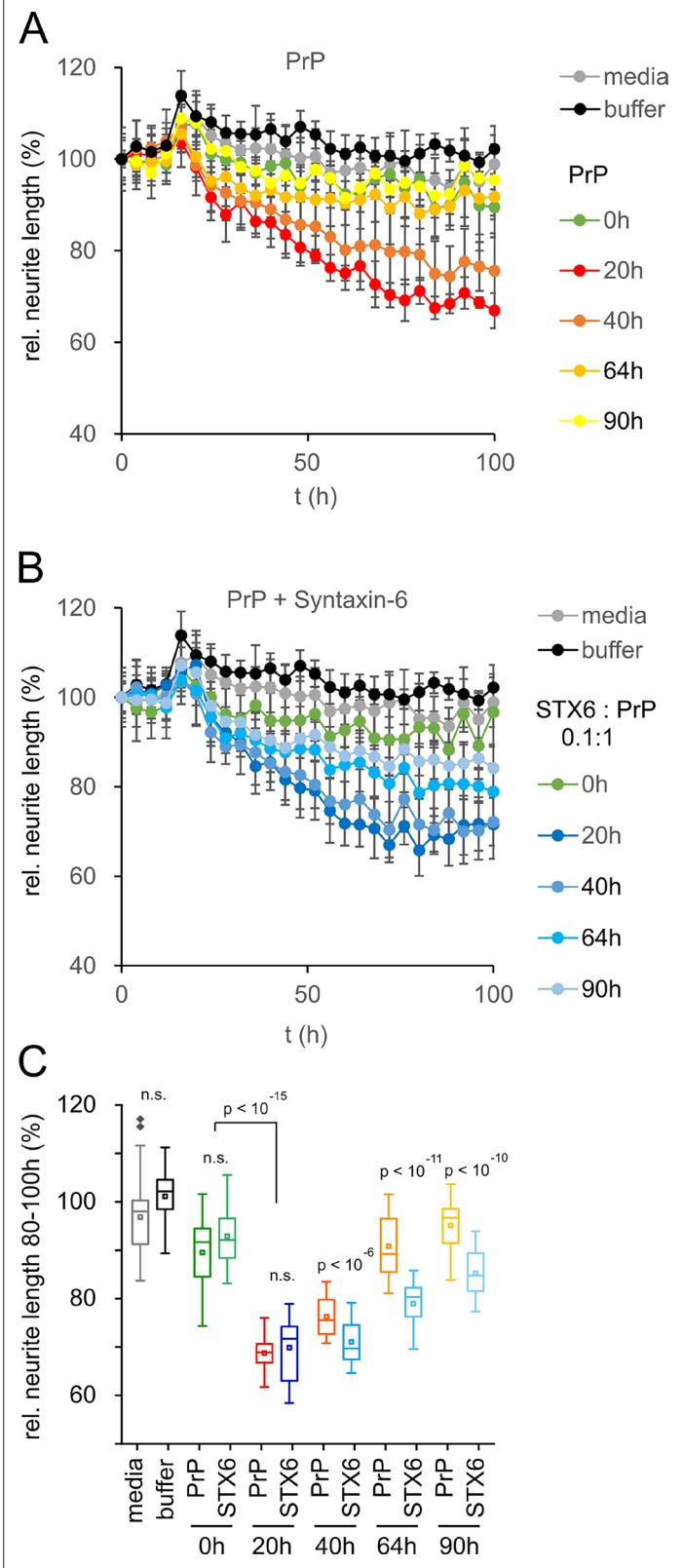

**Figure 5.** Toxicity in mouse primary neurons incubated with hPrP23/syntaxin-6 at different aggregation time points. Native aggregation assay (NAA) was performed as in *Figure 2* and samples were diluted 1:10 into cell culture media at the indicated time points (0 hr, 20 hr, 40 hr, 64 hr, 90 hr). (**A**) Plot of effect of PrP (250 nM) on neurite length compared to effect of media or NAA assay buffer at the same dilution from four independent wells. (**B**) Plot of

*Figure 5 continued*

effect of hPrP23 co-aggregated with syntaxin-6 (STX6, 1:10 molar ratio). (**C**) Box plot of average relative neurite lengths between 80–100 hr incubation of PrP in the cell culture media. p-Values were derived from ANOVA testing between PrP vs. PrP/syntaxin-6 samples at each time point.

The online version of this article includes the following figure supplement(s) for figure 5:

**Figure supplement 1.** Survival of primary neurons.

prion infectivity was reported for several assays (*Legname et al., 2004*; *Schmidt et al., 2015*), fibrils generally form under highly non-physiological conditions in the presence of high concentrations of guanidine or urea (*Legname et al., 2004*), at low pH (*Swietnicki et al., 2000*), in the presence of SDS (*Stöhr et al., 2008*), or at very high temperatures (*Atarashi et al., 2007*). None of these conditions provide a native environment for protein–protein interaction, and many cellular proteins will be unfolded under these assay conditions.

Full-length PrP formed fibrils with morphology and β-sheet secondary structure characteristic of amyloid under conditions of the NAA (*Willbold et al., 2021*). Fibril formation could be tracked by binding of the amyloidophilic dye ThT and displayed sigmoid aggregation kinetics typical of a nucleated polymerization mechanism (*Cohen et al., 2013*). As had been observed for other amyloidogenic proteins, solution conditions such as pH and temperature affected aggregation kinetics (*Buell et al., 2014*; *Milto et al., 2014*). Aggregation proceeded faster at neutral pH, while monomeric protein was stable at pH of 6.5 and below.

Seeding by preformed fibrils reduced the lag phase of aggregation (*Jarrett and Lansbury, 1993*; *Cohen et al., 2006*). The limiting dilution for seeding was ~$10^{-10}$ (w/w), which corresponds to approximately 2000 PrP molecules per well or 5 prion rods of a typical length of 200 nm (*Terry et al., 2016*), similar to other seeding assays (*Stöhr et al., 2008*; *Atarashi et al., 2007*; *Du et al., 2011*). In NAA, both lag phase $t_{50}$ and elongation rate constant $k$ displayed little dependence on monomer concentration, which suggests that monomer refolding was rate-limiting for fibril elongation. While in general aggregation kinetics scale with monomer concentration, elongation saturates at high monomer concentration as conformational change of PrP becomes rate-limiting (*Jain and Udgaonkar, 2008*; *Honda and Kuwata, 2017*). ThT fluorescence amplitudes scaled with PrP concentration between 2 and 16 µM, then dropped and lag phase increased. A competing non-fibrillar aggregation pathway at high protein concentration could account for this observation (*Powers and Powers, 2008*; *Bieschke et al., 2005*). High salt concentration shields charges, which promotes hydrophobic interactions and lowers the critical concentration of nucleated polymerization, often leading to the formation of amyloid fibril clusters (*Klement et al., 2007*). Notably, while PrP aggregated faster in the NAA at high salt concentration, it tended to form longer, more isolated fibrils.

## Inhibition of fibril formation by syntaxin-6

Syntaxin-6 delayed the formation of PrP fibrils at highly sub-stoichiometric ratios and co-aggregated with PrP into aggregate clusters. It delayed, but did not prevent, the formation of insoluble seeding-competent PrP assemblies. On the face, its effect was similar to that of the heat shock protein family A (HSP70) member 1A (*Figure 2*, *Figure 2—figure supplement 3*), which would suggest a role of syntaxin-6 as a potent aggregation inhibitor with chaperone-like activity. Chaperones assist folding of newly synthesized proteins, sequester soluble misfolded polypeptides, and target them for degradation, which makes them promising therapeutic targets in protein misfolding diseases (*Balch et al., 2008*; *Hartl, 1996*; *Muchowski and Wacker, 2005*).

HSP70 has been described as a 'holdase', which inhibits amyloid formation of various proteins at sub-stoichiometric concentrations even in the absence of ATP by binding to early-stage aggregation intermediates (*Evans et al., 2006*; *Wacker et al., 2004*). Overexpression or exogenous addition of HSP70 rescued cytotoxicity in cell cultures and model organisms (*Rosas et al., 2016*; *Bongiovanni et al., 2018*; *Fernandez-Funez et al., 2016*). Sequestering aggregation intermediates into inert off-pathway aggregates can be a powerful strategy against amyloid toxicity (*Bieschke et al., 2010*; *Ehrnhoefer et al., 2008*). However, our data indicate that, far from detoxifying PrP aggregation intermediates, the delay of PrP fibril formation by syntaxin-6 exacerbated PrP toxicity by allowing toxic aggregation intermediates to persist for longer (*Figure 5*).

Conversely, natural and chemical chaperones can detoxify aggregation intermediates by promoting fibril formation and stabilizing mature amyloid fibrils (*Cohen et al., 2006*; *Bieschke et al., 2012*; *Lam et al., 2016*). Syntaxin-6 seems to present the flip side of this benign activity in delaying PrP fibril formation and thus prolonging the presence of toxic aggregation intermediates.

The nature of the toxic species in prion disease is under active investigation. Brain homogenates from scrapie-infected mice have a specific toxic component, which can be blocked by the addition of anti-PrP antibody (*Benilova et al., 2020*). However, highly purified infectious prion rods were shown not to be directly neurotoxic (*Benilova et al., 2020*), suggesting that toxicity may be caused by non-prion assemblies, which accumulate after the infectious titer has peaked and which, unlike prions, are sensitive to sarkosyl (*Reilly et al., 2022*; *Sandberg et al., 2011*; *Sandberg et al., 2014*) or by transient aggregation intermediates, whose formation may be catalyzed by prions or other fibrillar PrP assemblies (*Collinge and Clarke, 2007*).

The chaperone Brichos breaks generation of toxic Aβ oligomers, which are formed through surface-catalyzed secondary nucleation (*Cohen et al., 2015*). Conceivably, syntaxin-6 could have the opposite effect on PrP fibril formation, in which toxic oligomer formation is promoted, by delaying competing fibril formation pathways. PrP itself binds to and inhibits the elongation of Aβ amyloid fibrils (*Amin and Harris, 2021*; *Bove-Fenderson et al., 2017*) oligomers and nanotubes. The presence of PrP during aggregation led to the formation of larger numbers of shorter fibrils and increased Aβ neurotoxicity in a dose-dependent manner (*Amin and Harris, 2021*). This suggests that PrP not only acts as a receptor for toxic amyloid oligomers (*Freir et al., 2011*; *Nicoll et al., 2013*), but may exacerbate neurotoxicity in a mechanism similar to syntaxin-6. Notably, syntaxin-6, when added to preformed fibrils, preferentially bound to fibril ends and other hotspots (*Figure 3*). Previous studies found only weak binding of syntaxin-6 to monomeric PrP$^C$ (*Jones et al., 2020*), which, together with the inhibition at sub-stoichiometric concentrations, strongly suggests an interaction with early aggregation intermediates and with fibrillary assemblies, possibly promoting secondary nucleation.

While variants at the *STX6* locus are known risk factors for sCJD (*Jones et al., 2020*), its deletion only modestly delayed the incubation period in RML prion infected mice, an observation that is open to different interpretations (*Hill, 2023*). Rather than directly altering prion replication kinetics, the protein may confer risk of disease by either facilitating the initial generation of prions in sporadic disease or exacerbating prion-associated toxicity. Our results lend support to the second hypothesis as syntaxin-6 interacted with early aggregation intermediates of PrP in vitro and exacerbated their toxicity (*Figure 5*) but did not accelerate the formation of seeding-competent PrP species (*Figure 2—figure supplement 3*) nor directly affected the replication of pre-existing prion seeds in PMCA reactions (*Figure 4B*). This finding argues against the hypothesis that syntaxin-6 binding to mature PrP fibrils induces fibril breakage. Correspondingly, an increase in fibrils branching/secondary nucleation as observed in the presence of syntaxin-6 (*Figure 3—figure supplements 1 and 3*) would have little effect on replication kinetics dominated by fragmentation/elongation.

## Localization of syntaxin-6 and PrP

A direct effect of syntaxin-6 on PrP assembly and/or prion-associated toxicity in vivo obviously requires the two proteins to be present in the same sub-cellular compartment. PrP$^C$ is present mainly in the TGN and on the outer plasma membrane, while mature PrP$^C$ is tethered to the membrane by a C-terminal glycophosphadityl-inositol anchor. At the plasma membrane, PrP$^C$ is incorporated into lipid rafts and caveolae, which are membrane microdomains enriched in cholesterol and sphingolipids. However, it has also been detected in multi-vesicular bodies and, to a small degree, in the cytosol (*Grassmann et al., 2013*). Both lipid rafts and the endocytic pathway are implicated as sites of prion replication (*Grassmann et al., 2013*).

Interestingly, syntaxin-6 has role in caveolin membrane recycling (*Choudhury et al., 2006*). As a SNARE protein, syntaxin-6 sits on the outside of vesicles/intracellular compartments and on the inner plasma membrane, so that a direct interaction with PrP would require mislocation of either protein. Both PrP and syntaxin-6 contain a trans-membrane domain and PrP can populate transmembrane forms in neurons (*Hegde et al., 1998*; *Stewart and Harris, 2005*). It is therefore plausible that, similar to the tau protein, PrP and prions bound to the membrane bilayer could interact with the transmembrane domain of syntaxin-6. Alternatively, mislocation of misfolded PrP assemblies could facilitate interaction with syntaxin-6. Amyloid assemblies can disrupt membrane integrity through mechanical

rupture, pore formation, or altering membrane curvature (*Shi et al., 2015*; *Lashuel et al., 2002*). Disruption of membranes on the cell surface and in vesicles of the endocytic pathway is a central step in the internalization of other misfolded polypeptides, such as Aβ (*Jin et al., 2016*; *Friedrich et al., 2010*). Similarly, it has been hypothesized to be necessary for cellular prion propagation (*Yim et al., 2015*). These processes could bring misfolded PrP into direct contact with syntaxin-6. FRET imaging suggests a direct contact between PrP and syntaxin-6 in a membrane-associated compartment in prion-infected cells (*Figure 4A*, *Figure 4—figure supplement 1B*), supporting this hypothesis.

## Conclusions

We developed an aggregation assay of murine and human prion protein under near-native conditions. PrP forms amyloid fibrils under the assay conditions, which bind ThT and can seed further fibril formation. Unlike natively unfolded proteins, however, PrP fibril formation does not scale strongly with monomer and seed concentrations, suggesting that structural conversion of native PrP into the amyloid fold is rate-limiting. This is consistent with the observation that the time constant of exponential prion replication in vivo is independent of PrP expression (*Sandberg et al., 2011*).

In the NAA, both the PrP and potential protein modulators of aggregation are in their native conformation, which means that the mechanisms by which protein and small modulators alter amyloid formation can be assessed. We analyzed the effect of syntaxin-6, a recently proposed risk factor for sCJD, on PrP self-assembly. To our surprise, we discovered that the protein acts as an 'anti-chaperone', which, by delaying PrP fibril formation, prolonged the persistence of toxic aggregation intermediates in vitro. Genetic variants in or near to *STX6* that enhance brain expression of the protein might therefore modify the risk of CJD by direct interaction with PrP and changing aggregation pathways, including the possibility of favoring more toxic aggregation intermediates.

While at this point we cannot assess whether the toxicity of PrP aggregation intermediates formed in vitro recapitulates the authentic PrP species responsible for neurotoxicity in prion disease in vivo, our data highlight a new mechanism by which protein modulators of amyloid formation can have counter-intuitive deleterious effects.

## Materials and methods

### Protein expression and purification

#### Human PrP (23-231), mouse PrP (23-231)

The open-reading frame of the human PrP gene (*PRNP*) (residues 23-231), containing methionine at residue 129 and the mouse PrP gene (*Prnp*) (residues 23-231, including S231), was synthesized de novo by Eurofins MWG Operon, with a thrombin-cleavable His-Tag added to the PrP N-terminus. The ligated pTrcHisB/PRNP and pTrcHisB/Prnp constructs were used to transform the *Escherichia coli* host strain BL21(DE3) (Novagen), genotype F′ ompT hsdSB (rB- mB-) gal dcm (DE3), which was then plated onto Luria-Bertoni (LB) agar plates containing 100 µg/ml carbenicillin. Cultures were grown for purification using a modification of protocols previously described (*Hosszu et al., 2005*). Briefly, following harvesting, cells were sonicated and their inclusion bodies containing PrP resolubilized in 6 M guanidine hydrochloride (GuHCl), 50 mM Tris-HCl, 0.8% β-mercaptoethanol, pH 8.0. These were loaded onto a Ni-NTA column equilibrated in 6 M GuHCl, 10 mM Tris-HCl, 100 mM $Na_2PO_4$, 10 mM glutathione pH 8.0, and eluted from the column using 10 mM Tris-HCl, 100 mM $Na_2PO_4$, 2 M imidazole pH 5.8. Residual GuHCl was removed through dialysis against 20 mM Bis-Tris.HCl pH 6.5, $CaCl_2$ added to a final concentration of 2.5 mM, and the N-terminal His-tag cleaved by thrombin for 16 hr at room temperature (RT) (0.1 U thrombin [Novogene]/1 mg of PrP added). The cleaved protein was loaded onto a second Ni-NTA column equilibrated with 20 mM Bis-Tris pH 6.5, 25 mM imidazole pH 6.5, and the eluted PrP peak was collected and dialyzed against 10 mM Bis-Tris pH 6.5, and aliquoted and stored at –80°C. Protein concentrations were determined by UV absorption at 280 nm using a calculated molar extinction of 56,667 $M^{-1}$ $cm^{-1}$ and 62,268 $M^{-1}$ $cm^{-1}$ for human and mouse PrP, respectively (https://web.expasy.org/protparam).

#### SEC of syntaxin-6/hPrP23/mPrP23

2 ml pre-concentrated, pre-cleared protein was loaded via injection valve onto a Sephacryl S100HR column (GE Healthcare) (26/60: 320 ml bed volume) that had been pre-equilibrated with 20 mM

Tris-HCl, 0.2 M NaCl, pH 8. A flow rate of 2 ml/min was used; all eluted peaks were checked on a silver-stained NuPage Bis-Tris 12% gel, using the protocol according to the silver staining kit (Silver-Quest Silver Staining Kit-1, LC6070, Life Technologies Ltd) and the purified protein was aliquoted and stored at −80°C until use.

## Stathmin 1 (STMN1)/heat-shock protein 70 (HSPA1A)

Glycerol stocks of *E. coli* BL21 strains expressing His-tagged STMN1 and HSPA1A were a gift by E. Wanker, MDC-Berlin, Germany. Small-scale cultures of STMN1 and HSPA1A were cultured in LB/Amp (100 µg/ml) by inoculating with stabs from glycerol stocks. Cells were pelleted and then inclusion bodies lysed in 1 ml lysis buffer (containing 50 mM Tris, 0.2 M NaCl, pH 8 containing 0.1% Tween20, 0.5% NP40, 50 U/ml benzonase, 10 µg/ml lysozyme, and 1 mM PMSF) placed on ice for 1 hr with gentle vortexing every 15 min. (All centrifugations were done at 4°C to further prevent non-specific cleavage by proteases and samples were kept on ice.) The lysate was cleared by centrifugation (1 hr, 16,100 rpm) and then loaded onto 2 ml NiNTA resin bed packed in mini spin columns, pre-equilibrated with buffer containing 50 mM $NaH_2PO_4$, 0.3 M NaCl, 20 mM imidazole at pH 8. The cleared lysate was loaded onto the column, unbound was eluted off the column first with gentle centrifugation (2 min, 2000 rpm); the resin was washed to thoroughly remove unbound protein by twice loading 700 µl pre-equilibration buffer (see above) and centrifugation for 30 s, 2000 rpm, and then 2 min, 2000 rpm for the second wash. Finally, the pure protein was eluted in elution buffer (containing 50 mM $NaH_2PO_4$, 0.5 M NaCl, 0.5 M imidazole, pH 8). This was done by loading 100 µl elution buffer with a 30 s centrifugation at 2000 rpm followed by a second elution with 150 µl elution buffer and a 2 min centrifugation at 2000 rpm. Both fractions were pooled and checked for purity as below.

The purified STMN1 and HSP70 recombinant proteins were checked for purity by Coomassie staining after PAGE on NuPage Bis-Tris 12% gels. Protein identity was confirmed by mass spectrometry following a standard trypsinization protocol on a Waters Xevo-XS spectrometer as described in *Manka et al., 2022*.

## Syntaxin-6 (STX6)

Syntaxin-6 (residues 38-318;accession number KU144824) was prepared according to *Jackson et al., 1999* with modifications. Briefly, the DNA sequences encoding the syntaxin-6 protein in pQTEV were transformed into BL21 (DE3). BL21 cultures were grown in LB medium in the presence of 100 µg/ml ampicillin. Expression of the protein was induced using 1 mM IPTG and was purified from inclusion bodies under denaturing conditions using nickel superflow resin with an AKTA Pure (GE Healthcare Life Sciences). The protein was refolded on NiNTA resin and eluted from the column using an imidazole gradient. The eluted material was extensively dialyzed against 20 mM Tris, 2 mM EDTA, 10 mM DTT, 200 mM NaCl pH 8.0. DTT concentration was reduced to 2 mM in the final storage buffer. Syntaxin-6 was further purified by size-exclusion chromatography as detailed above, resulting in a protein band with an apparent molecular weight of ~28 kD (*Figure 1—figure supplement 1A*). The final concentration of the syntaxin-6 protein was determined by absorption measurement at 280 nm, $\varepsilon$ = 31,970/M/cm. Aliquots were stored at −80°C until use.

## Circular dichroism (CD) spectroscopy

CD data were recorded on a Jasco J-715 spectrophotometer equipped with a thermoelectric temperature controller. A 1 mm path length cuvette was used for all CD spectroscopy measurements. Wavelength measurements were recorded from 195 nm to 260 nm at 20°C using 0.36 mg/ml syntaxin-6 in 10 mM Na-phosphate, pH 8.

## Native aggregation assay (NAA)

PrP$^C$ was filtered through a 100 kD membrane filter in 10 mM Bis-Tris pH 6.5 (Amicon ULTRA 0.5 ml 100K 96PK, UFC510096, Merck Life Science UK Ltd) to remove aggregates and then diluted into reaction buffer (50 mM Na-phosphate, pH 6.8, 150 mM NaCl, 0.01% Na-Azide, 20 µM ThT and 5 mM Bis-Tris pH 6.5) with the addition of 0.1% seed (w/w), unless indicated otherwise. The standard final concentrations of PrP and NaCl were 2.5 µM and 150 mM, pH 6.8 unless indicated otherwise. The seed consisted of aggregated PrP material from an unseeded reaction under the same conditions as above.

It was diluted to 10% (w/w) in 5 mM Bis-Tris, pH 6.5 and sonicated in a water bath sonicator (GRANT, XUBA1) for 15 min prior to addition to the reaction mix. Aggregation assays for neuronal toxicity were set up in parallel to the standard assays for kinetic analysis, but here the ThT and Na-Azide was omitted from the reaction mixes.

The seed and post-aggregation samples were handled with pre-silanized tips, in pre-silanized non-binding tubes (Repel-silane ES, Sigma, 17-1332-01). The reaction mix (94 µl per well) was dispensed into low binding 96-well COSTAR (#3651) plates and three silanized zirconium beads (0.5 mm diameter, *Figures 1–4*; 1 mm diameter, *Figure 5*) were added to each well. A single glass bead (2 mm) was added instead of the Zr beads where indicated. ThT kinetics were recorded on a BMG ClarioStar plate reader at 42°C with a shaking speed of 700 rpm, set to shake for 100 s with an incubation time of 20 s between agitation. Focal height was set to 20 mm, top read, with excitation at 440 nm and emission set to 485 nm. Aggregation kinetics were fitted in MATLAB E2021b using the following equation:

$$F = f_0 + \frac{A}{1 + e^{-k(t - t_{50})}}$$

(1)

$$t_{lag} = t_{50} - \frac{2}{k}$$

For Amylofit analysis (*Meisl et al., 2016*), kinetic traces were normalized to the first 5 hr of the plateau phase and noise in the fluorescence signal was smoothed by calculating a 6 hr moving average. Half-times were calculated in Amylofit and kinetics were analyzed at monomer concentrations of 2, 6, 10, 16, and 20 µM and a seed concentration of 2 nM (monomer equivalent) using the models for fragmentation, saturated elongation + fragmentation, secondary nucleation + fragmentation, and nucleated polymerization.

## EM grid preparation

Endpoint samples (5–6 µl) from NAAs sonicated in a water bath for 10–15 s were loaded onto carbon-coated 300 mesh copper grids (Electron Microscopy Sciences) that had been glow-discharged for 40 s using an PELCO easiGLOW glow discharge unit (Ted Pella Inc, USA). Samples were left to bind for 30 min, blotted dry, washed in water (1 × 50 µl), blotted, and then stained with 10 µl Nano-W (methylamine tungstate) stain (Nanoprobes) for 1 min followed by 30 s (with blotting in between stain times). Images were acquired on a Talos electron microscope (FEI, Eindhoven, NL, now Thermo Fisher).

## Immunogold labeling of PrP fibrils with NAPTA precipitation

40 µl of fibrils (completion at 115 hr) were dispensed into an RNase free pre-silanized tube with silanized tips and centrifuged for 1.5 hr (4°C, 16,100 rpm) after which the pellet was resuspended in 500 µl TBS containing 0.1% (w/v) sarkosyl with sonication for 30 s in a water bath sonicator (GRANT, XUBA1), and incubated with syntaxin-6 antibody (C34B2) Rabbit mAb 2869 (Cell Signaling Technology Europe B.V, 2869S; 1:100) for 16 hr at 25°C with gentle agitation.

The following day, the fibrils were precipitated with sodium phospho-tungstate (NaPTA) by addition of 40.5 µl of 4% (w/v) NaPTA (prepared in H$_2$O; pH 7.4) and centrifuged at top speed for 30 min to recover a pellet which was resuspended in 10 µl total volume of 1:20 (v/v) Goat anti-Rabbit IgG conjugated to colloidal gold (10 nm) (Insight Biotechnology Ltd, GA1014) in TBS containing 5% (v/v) glycerol and incubated at 25°C for 3 hr with gentle agitation. The sample was pulse centrifuged for 5 s and 5 µl labeled sample was loaded onto a glow-discharged carbon-coated grids and stained with Nano-W as previously described without the 10–15 s pre-sonication step.

## SR sample preparation

Eight-well chamber slides (IBL Baustoff, 2 20.140.082, C8-1.5-H-N, 8-well chambered cover glass with #1.5 high-performance cover glass, 0.170 ± 0.005 mm) were cleaned by soaking them overnight in a 2% solution of Hellmanex II detergent diluted in ultrapure water and then washed thoroughly in ultrapure water before rinsing in 100% methanol and then a final thorough rinse in ultrapure water and then allowing the slides to dry.

Once dry, the slides were glow-discharged for 40 s using a PELCO easiGLOW glow discharge unit (Ted Pella Inc). 10 µl sample was added to the center of a well and allowed to incubate for 1 hr for the

fibrils to adhere to the glass. The well was then washed 5× with 500 µl of HPLC purified water before 200 µl buffer (which consists of 10 nM Nile Red; Sigma-Aldrich, 72485-1G in PBS or GLOX buffer). The wells were then sealed either with Nescofilm or TWINSIL before imaging on a custom-built TAB/dSTROM super-resolution microscope (Sun et al., 2023).

Enzymatic oxygen scavenger (GLOX, glucose oxidase with catalase) buffer consisted of two solutions. Solution A: Tris (50 mM, pH 8.3), NaCl (10 mM), glucose (10% w/v), and β-mercaptoethylamine (Sigma-Aldrich, 30070, 10 mM). Solution B: glucose oxidase (Sigma-Aldrich, G2133, 8 mg), and catalase (Sigma-Aldrich, C100, 38 µl, 21 mg/ml) in PBS (160 µl). Solutions A and B were mixed at the ratio of 99:1 (v/v) immediately before use.

## Labeling recombinant protein with AlexaFluor dyes

AlexaFluor dyes (AlexaFluor 488 NHS Ester, A20000, AlexaFluor 647 NHS Ester, A20006, AlexaFluor 488 C5 Maleimide, A10254, https://www.thermofisher.com) were diluted in DMSO to a stock concentration of 10 mg/ml. The dyes were mixed with the recombinant protein constructs at a molar ratio of 2:1 dye:protein construct. The mixture of dye and protein constructs was protected from light by covering the vessel with foil and left to mix on a rotator overnight at 4°C. The following day, the unbound dye was removed by dialysis in 2.5 l for 2 × 1 hr, with 8 kD dialysis membrane (SpectraPor 7 8000 Dalton MWCO; 11425919, Fisher Scientific, UK). For syntaxin-6 the buffer used for dialysis was 10 mM Na-phosphate, 150 mM NaCl, 2 mM EDTA, 2 mM DTT, pH 8; for PrP, the buffer used for dialysis was 10 mM Bis-Tris, pH 6.5. Once dialysis was complete, the labeled protein was recovered from the dialysis membrane and checked on a 4–12% Bis-Tris gel. To check labeling efficiency, an absorption spectrum measurement (200–700 nm) was taken and the final concentrations of dye and protein were calculated according to the manufacturer's protocol. The labeled proteins were then aliquoted and stored at –80°C.

## Primary neuronal culture and toxicity analysis

Primary neuronal cultures were derived from brains of unmodified inbred FVB/N mice. Hippocampi of male and female E17 mouse brains from a single litter (7–9 embryos) were dissected in HBSS (Thermo Fisher Scientific) supplemented with 1% L-glutamine, 1% HEPES, and 1% pen-strep. Cells were dissociated using 0.25% trypsin + 0.04% benzonase, triturated mechanically, and counted using a Neubauer haemocytometer. Cells were plated in DMEM supplemented with 10% horse serum (#26050-88, Invitrogen) at 10K/well to the inner 60 wells of poly-L-lysine-coated 96-well plates (Greiner, 655936). At 1 hr post-plating, DMEM was aspirated and exchanged for Neuralbasal medium (21103049, Thermo Fisher Scientific) supplemented with 0.25% GlutaMAX (35050061, Thermo Fisher Scientific) 2% Gibco B27 supplement (17504044, Thermo Fisher Scientific) and incubated at 37°C (20% $O_2$, 5% $CO_2$). FVB/N neurons were maintained in culture for 11–12 days prior to a 96 hr treatment. Images of live cells were taken on an IncuCyte S3 reader (Sartorius) with a ×20 objective in phase contrast. Four views were captured in each well every 4 hr. PrP samples were diluted 1:10 into fresh media and added at the 24 hr mark.

Neurite lengths were evaluated using the NeuroTrack module of the IncuCyte S3 software package (rev 2019A) using the following parameters: cell body cluster segmentation 0.7; cleanup 0; cluster filter 0; neurite sensitivity 0.25; and neurite width 1 µm. Detected neurite masks are highlighted in pink in images. Neurite length data were normalized to the initial (0 hr) value for each well and means ± standard deviation were calculated from quintuplicate sample wells. Average neurite lengths at the 80–100 hr time interval were visualized in a box plot.

The number of live neurons was counted after 0- and 4-day incubation as a secondary toxicity assay. Data represent averages from four images under each condition and time point, normalized to the number of neurons present at the time of protein addition in each field of view. ANOVA statistical analysis was performed in OriginPro 2019.

## FRET imaging

500 µl of 50,000 cells/ml (uninfected S7 and infected iS7 subclones of PK1 N2a neuroblastoma cells) were plated in eight-well chambered glass coverslips (Thermo Scientific [155411]) and incubated at 37°C/5% $CO_2$ for 3 days. Cells were then fixed for 15 min at RT with 3.7% formaldehyde diluted in Dulbecco's phosphate-buffered saline (DPBS; Gibco [14190-094]). After washing once with DPBS, cells

were treated with ice-cold acetone for 1 min. Cells were washed before treatment with 3.5 M GdnSCN for 10 min. Following five washes, the cells were incubated with anti-syntaxin-6 (Cell Signaling Technologies [#2869], clone C34B2, 1:300) and/or anti-PrP (BioLegend [808001] clone 6D11, 1:10,000; Santa Cruz [sc-47730], 5B2, 1:500; Sigma [P0110], clone 8H4, 1:500) in sterile-filtered 0.25× SuperBlock in PBS overnight at 4°C, followed by secondary antibodies (AlexaFluor 647-AffiniPure Goat Anti-Rabbit IgG (H+L) and/or Rhodamine Red-X (RRX) AffiniPure Goat Anti-Mouse IgG (H+L), 1:1000) and a DNA counterstain (DAPI; 1:10,000) in 0.25× SuperBlock overnight at 4°C. Antibodies were removed, and following one wash imaging was performed with a Zeiss LSM 710 laser-scanning microscope with oil-immersion ×63/1.4 NA objective.

Four images were acquired for each sample: (a) donor (excitation 561 nm, emission 580–610 nm); (b) acceptor (633 nm excitation, 670–800 nm emission); (c) FRET excitation 561 nm, emission 670–800 nm, all using the MBS 488/561/633 triple dichroic mirror; and (d) DAPI (405 nm excitation, 440–460 nm emission). Laser power, photomultiplier gain, and pinhole size were adjusted as to not exceed the PMT dynamic range. Identical parameters were used for all image acquisition. Three biological replicates were imaged, with n = 3–6 images each.

FRET data were analyzed using the ImageJ PixFRET plugin according to the developer's manual (*Feige et al., 2005*). Syntaxin-6-only-stained iS7 cells were used as acceptor bleed-though control and 5B2/6D11/8H4-only stained cells were used as donor bleed-through controls, respectively. FRET images were calculated at 1 pixel Gaussian blur and rendered using the Parula HDR lookup table at a range of 0–500.

## Cell lines
### Neuro2a (N2a) cell line and derivatives
N2a cells (male donor) were sourced from ATCC (CCL-131). Prion susceptible subclones of N2a cells were derived as described in *Klöhn et al., 2003*; *Marbiah et al., 2014* and authenticated by transcriptomic analysis, where gene expression differences between original N2a cells and prion-susceptible subclones (PK1 S7, iS7) were investigated and documented (*Marbiah et al., 2014*). All actively used N2a sublines are tested for mycoplasma contamination at least once every 2 years. No mycoplasma contaminations have been reported in the MRC Prion Unit in the last 10 years.

## PMCA using *Stx6*$^{+/+}$ and *Stx6*$^{-/-}$ mouse brain homogenate
### Tissue collection
Brains for the PMCA substrate were derived from juvenile *Stx6*$^{+/+}$ and *Stx6*$^{-/-}$ male C57BL/6N mice at 3 months of age. Animals were sacrificed by $CO_2$ asphyxiation and immediately perfused with 20 ml ice-cold perfusion buffer (1× DPBS + 5 mM EDTA). The perfused brain was then removed and stored frozen at –70°C until use.

### Substrate preparation
9% (w/v) mouse brain homogenates were prepared in ice-cold conversion buffer (1× DPBS, 150 mM NaCl, 1% [v/v] Triton X-100) with protease inhibitors (1x Protease Complete with EDTA) with a dounce, glass homogenizer. Debris was removed by centrifuging the homogenates at 1200 × *g* for 2 min, with the supernatant subsequently being collected and stored at –70°C until use.

### PMCA
0.8 µl of seed (I6200 10% [w/v] brain homogenate from terminal RML-infected CD-1 mice) was spiked into 0.2 ml PCR tubes containing 79.2 µl of *Stx6*$^{+/+}$ or *Stx6*$^{-/-}$ substrate containing three 0.5-mm-diameter zirconium beads. Samples were subjected to PMCA in an automated sonication bath (QSONICA) at 35°C for 96 cycles (50% amplitude, 30 s sonication every 30 min) for a total of 48 min sonication over 48 hr. Following sonication, the samples were briefly centrifuged before storage at –80°C. Control reactions were also prepared, which were directly frozen.

### Immunoblotting
Immunoblotting was performed as previously described with minor modifications. Briefly, each reaction was incubated with proteinase K (Roche, Cat# 3115887001) at 50 µg/ml for 30 min at 37°C. 20 µl

of PK-digested material was electrophoresed and gels electroblotted. Membranes were probed with 200 ng/ml ICSM35 anti-PrP antibody (D-Gen Ltd) in PBST for 1 hr at RT or overnight at 4°C. After washing, the membranes were probed with a 1:10,000 dilution of alkaline-phosphatase-conjugated goat anti-mouse IgG secondary antibody (Sigma-Aldrich, A2179) in PBST. Blots were incubated for 5 min with CDP-Star Chemiluminescent Substrate (Thermo Scientific, T2147) and visualized on Biomax MR film (Kodak) or visualized on a LiCor Odyssey imager using anti-mouse pAb-IRdye800CW (1:5000) as secondary antibody.

RML prion rods for immunoblotting were prepared as described in *Wenborn et al., 2015*. Endpoint NAA samples (160 µl, 2.5 µM hPrP23 monomer equivalent) were digested with proteinase K (50 µg/ml) at 37°C for 30 min, pelleted with 4% (w/v) NaPTA as described above, resuspended in 10 µl water, and sonicated for 15 min in a water bath sonicator (GRANT, XUBA1) before SDS-PAGE and blotting.

Work with animals was performed under the license granted by the UK Home Office (Project Licenses 70/6454 and 70/7274) and conformed to University College London institutional and ARRIVE guidelines.

## Acknowledgements

We thank Adam Wenborn and Dr. Jonathan Wadsworth, MRC Prion Unit, for providing mouse scrapie material and mass spectrometry; Dr. Peter Klöhn for providing PK1 S7 and iS7 cell lines; Prof. Erich Wanker, MDC-Berlin, Germany, for providing *E. coli* clones for protein expression; and the staff of the MRC Prion Unit at UCL Biological Services Facility for animal husbandry and care. We thank Dr. Georg Meisl, University of Cambridge, for helpful discussions. The research was supported by the National Institute of Neurological Disorders and Stroke of the National Institutes of Health grant number 1R21NS101588-01A1, MRC grant MC_UU_00024/6 to JB, and the MRC Prion Unit at UCL graduate programme. The Stx6⁻/⁻ mice were obtained from the MRC Harwell Institute, which distributes these mice on behalf of the European Mouse Mutant Archive (https://www.infrafrontier.eu/emma/). The MRC Harwell Institute is also a member of the International Mouse Phenotyping Consortium (IMPC) and has received funding from the Medical Research Council for generating and/or phenotyping the Stx6⁻/⁻ mice. The research reported in this publication is solely the responsibility of the authors and does not necessarily represent the official views of the Medical Research Council. Funding and associated primary phenotypic information may becan be found at https://www.mousephenotype.org.

## Additional information

### Funding

| Funder | Grant reference number | Author |
|---|---|---|
| National Institute of Neurological Disorders and Stroke | 1R21NS101588-01A1 | Jan Bieschke |
| Medical Research Council | MC_UU_00024/6 | Daljit Sangar<br>Mark Batchelor<br>Graham S Jackson<br>Jan Bieschke |
| Medical Research Council | MRC Prion Unit Graduate Programme | Elizabeth Hill<br>Kezia Jack<br>Beenaben Mistry |

The funders had no role in study design, data collection and interpretation, or the decision to submit the work for publication.

### Author contributions

Daljit Sangar, Kezia Jack, Formal analysis, Investigation, Writing - review and editing; Elizabeth Hill, Formal analysis, Investigation, Methodology, Writing - review and editing; Mark Batchelor, Formal analysis, Investigation; Beenaben Mistry, Investigation; Juan M Ribes, Resources, Methodology; Graham S Jackson, Formal analysis, Supervision, Investigation, Methodology, Writing - review and

editing; Simon Mead, Supervision, Writing - review and editing; Jan Bieschke, Conceptualization, Formal analysis, Funding acquisition, Writing - original draft, Writing - review and editing

### Author ORCIDs
Mark Batchelor ⓘ https://orcid.org/0000-0001-6847-5131
Jan Bieschke ⓘ https://orcid.org/0000-0002-3485-9767

### Ethics
Work with animals was performed under the licence granted by the UK Home Office (Project Licences 70/6454 and 70/7274) and conformed to University College London institutional and ARRIVE guidelines.

### Decision letter and Author response
Decision letter https://doi.org/10.7554/eLife.83320.sa1
Author response https://doi.org/10.7554/eLife.83320.sa2

---

## Additional files

### Supplementary files
• Reporting standard 1. ARRIVE E10 checklist.

• MDAR checklist

### Data availability
All data generated or analysed during this study and all used analysis scripts have been uploaded to Mendeley data and are available to the public under https://doi.org/10.17632/yggpkrgnx8.1.

The following dataset was generated:

| Author(s) | Year | Dataset title | Dataset URL | Database and Identifier |
|---|---|---|---|---|
| Bieschke J | 2024 | Syntaxin 6 delays prion protein fibril formation and prolongs presence of toxic aggregation intermediates | https://doi.org/10.17632/yggpkrgnx8.1 | Mendeley Data, 10.17632/yggpkrgnx8.1 |

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
