## [Editor Report]

The current study presents an important discovery about Syntaxin 6 (Stx6)'s anti-chaperoning activity on PrP. The authors provide compelling evidence that the anti-chaperone activity arises as Stx6 delays PrP fibril formation and in the presence of Syntaxin 6, the amorphous aggregates of PrP are more toxic to neuronal cells. This study provides a critical molecular link between PrP aggregation and neurotoxicity.

---

## [Decision Letter]

**Decision letter after peer review:**

[Editors’ note: the authors submitted for reconsideration following the decision after peer review. What follows is the decision letter after the first round of review.]

Thank you for submitting the paper "Syntaxin 6 delays prion protein fibril formation and prolongs presence of toxic aggregation intermediates" for consideration by *eLife*. Your article has been reviewed by 3 peer reviewers, one of whom is a member of our Board of Reviewing Editors, and the evaluation has been overseen by a Senior Editor. The following individuals involved in review of your submission have agreed to reveal their identity: Kausik Chakraborty (Reviewer #2); Samir K Maji (Reviewer #3).

Comments to the Authors:

We are sorry to say that, after consultation with the reviewers, we have decided that this work will not be considered further for publication by *eLife*. As you will see below, reviewers 1 and 2 were critical with respect to the physiological relevance of interaction between Stx6 and PrP in addition to raising several other technical points. Reviewer 3 also asked many pertinent questions, as elaborated below. The time that it would take to satisfactorily address these concerns may be substantial, which was the basis for declining the work in its current form.

*Reviewer #1 (Recommendations for the authors):*

The suggested experiments can be addressed by the authors to make their claim stronger:

1) Authors should check in vivo interaction of PrP and Stx6 in different conditions that favors prion pathogenesis. If there is no interaction between Stx6 and Prp, the whole hypothesis of Stx6-mediated enhancement of prion pathogenesis would be baseless.

2) Figure 2D: What is the assay end point? Is it 115h? If it is 115h, then figure 2E should also show the status of aggregates at the same time point rather than at 65h.

3) Figure 5: The reduction in relative neurite length by Prp-Stx6 aggregates compared to PrP fibrils as an indicator of neuro-toxicity although significant but very small in % change. Is this much of change in neurite length with Stx6-PrP aggregates justify the enhanced toxicity model by Stx6? Can authors use some other assays like cell death assays that can show the enhanced toxicity more clearly?

*Reviewer #2 (Recommendations for the authors):*

In this paper, Sangar.et al. discuss an in vitro assay to generate toxic PrP aggregates under near native conditions using purified human and murine PrP (Prpc) which allows them to characterize protein-amyloid interactions. Interestingly, they find that interaction of STX6 (Syntaxin-6) leads to formation of amorphous PrP aggregates instead of fibrillar aggregates which prolongs toxicity in primary neurons. There is novelty in the work but has not been sufficiently explored.

Details:

1) Does Stx6 bind to preformed prions, or do they bind to the prions only while they are forming? If they do bind to preformed prions their effect on aggregation kinetics may have a different explanation (please see the comment below).

2) Stx6 seems to only increase the lag phase (with a minor effect on the elongation rate) – does it interfere with the nucleation kinetics by binding to the seeds, or do they bind to the folded/unfolded PrP to change the equilibrium and hence alter the nucleation kinetics?

Without understanding these basics, it is hard to conclusively state the mechanism of Stx6's effect on PrP aggregation.

3) Is stx6 secreted when they are overexpressed? Stx6 overexpression is a risk factor for CJD, so how is the in vivo expression level related to prion formation?

4) How do the authors connect the expression level correlation to PrP toxicity to formation of PrP/Stx6 mixed aggregates?

5) Is there any evidence that PrP/Stx6 forms mixed amyloids in vivo?

6) How do the authors deconvolute the toxicity of PrP and Stx6 in the PrP/Stx6 mixed amyloids? The toxicity could be due to Stx6 or PrP aggregates/amyloids?

7) The toxicity differences are very modest given stark differences in aggregate morphology in the presence or absence of Stx6. How do the authors explain this?

Other comments:

1) Apart from the development of an in vitro setup to generate toxic PrP aggregates, a major finding of the manuscript is the interaction between STX6 and PrP which prevents formation of fibrillar aggregates supported by ThT fluorescence assay, immunogold-EM analysis, and high-resolution microscopy and cytotoxicity data using primary neurons. From the data shown here we can convincingly conclude the following:

a) Recombinant PrP forms fibrillar aggregates in the assay conditions described.

b) STX6 and PrP interact in vitro.

c) This interaction prevents formation of fibrillar aggregates.

2) However, the data to show that STX6 interacts with PrP in-vivo (both in its monomeric and aggregated form) is lacking. The co-localization of Syntaxin-6 in the PrP aggregates isolated from clinical samples (patient samples) would be a crucial validation of the observations made in-vitro i.e, Interaction of Syntaxin-6 with PrP aggregates. While this may be complicated, it will be difficult to interpret the observations made in vitro in the absence of in vivo data.

3) And more importantly, the article could be further strengthened by more robust biochemical analysis into the mechanism of how STX6 interacts with PrP.

4) Following up on point 3, the authors may investigate whether STX6 interaction facilitates release of PrP from cells similar to tau (Lee et al. 2021).

5) In the TEM micrographs, it is very hard to locate the aggregates (Figure 2F and S4B). But for the ease of understanding of general reader, it is important to show presence of both the aggregate and the colocalization of STX6 in the immunogold staining.

6) Does treatment of primary neurons with 1:1 stoichiometry of STX6 and PrP prevent PrP induced neurotoxicity? Concentration dependent analysis may be done. This will further validate the finding observed in Figure S2(A).

7) While this may be out of scope of the study, authors may look into the effects of overexpression or knockdown of syntaxin-6 on PrP aggregate formation.

*Reviewer #3 (Recommendations for the authors):*

The objectives of the project are well-addressed and the discussion part is nicely illustrated. Some of the results look very interesting, leading to some exciting predictions (e.g., refolding of monomer, formation of off-pathway intermediates, etc.) based on the initial result. However, some additional experiments are required for a better understanding of the present results. Hereby, the authors are requested to address the following concerns and perform additional experiments that are mentioned in the following section.

Major points:

• Although the fibrillation assay used in this manuscript does not use denaturant or detergent, however, the PrP proteins are kept at 42{degree sign}C, pH 6.8, under agitation with zirconium 86 beads. How a condition like agitation can be claimed as a native environmental condition and will ensure that the assay condition is not facilitating protein unfolding, unlike the other reported fibrillation assays?

• In line 121, the authors wrote that "formation of aggregates with more amorphous morphology, which seems to consist of short fibrils" this is inconsistent comment as proteins will form amorphous or amyloids. Amorphous protein aggregates can't compose of short fibrils?

• The Thioflavin T aggregation profile looks extremely noisy (e.g., Figure 1A, 2A) with huge fluctuation in the Y axis. The authors should explain the reason for this much fluctuation in the ThT signal and fit the kinetic profile taking care of the standard deviation.

• The authors have calculated lag times (t50) and fibril elongation rate (k) in different experimental conditions throughout the manuscript. However, the authors did not show the calculation performed to determine these two parameters in the method section. The authors should incorporate that. Moreover, the authors should perform a web-based global fitting software interface, such as Amylofit (Meisl et al., 2016), to precisely evaluate different kinetic parameters for better comparison.

• The quality of TEM images of hPrP (Figure 1C) and moPrP (Figure S2C) is of poor quality. The authors are requested to change these with high-resolution images. Moreover, the present immune-gold staining image (Figure 2F) does not show the fibril where immune-gold labeled Stx6 is incorporated. The authors should provide a better-quality immune-gold staining image with a clear fibril image so that the location of incorporated Stx6 (immune-gold labeled) is clearly understood.

• The authors have shown that full-length PrP proteins form amyloid fibrils under near-native conditions. The authors should centrifuge to isolate the fibril fraction and show the proteinase-K digestion of the fibril, which is a known property of PrP amyloid fibril. The authors also should compare this PK resistance data with native PrP amyloid isolated from mouse prion strains ME7 and RML that the authors have used. Further, the authors should also perform the same with the fibril bound to Stx6, Hspa1a and Stathmin 1 to understand the nature of the aggregates in the presence of the additives.

• The authors have speculated the formation of the "off-pathway" intermediate based on the observation that the lag phase of aggregation has increased and ThT amplitude has dropped at high monomer concentration. The author should isolate the intermediate through centrifugation at the particular aggregation condition where the predicted off-pathway intermediate populates the most. To precisely comment on the "off-pathway" intermediate, the authors should add the isolated intermediate fraction during the fibrillation kinetics in a dose-dependent manner to see the concentration effect of the intermediate in fibrillation kinetics that will provide direct evidence to support the statement.

• To check the interaction with PrP protein in super-resolution microscopy, the authors have chosen only Stx6 among HSPA1A, STMN1, and STX6. The authors should justify the specific reasons behind choosing Stx6 over the others.

• The authors have observed that the interaction sites of Stx6 were often located at fibril ends or kinks in the fibril. Thus, the authors have proposed a potential role of Stx6 in fibril breakage or secondary nucleation. Firstly, the authors should reconfirm the Stx6 binding location in PrP from the clear immune-gold staining images. Secondly, the authors should isolate the Stx6-bound fibrils and perform a seeding reaction (prion amplification) with only the PrP monomer to see whether Stx6-bound PrP fibrils are competent for elongation or not. If Stx6 binds to the fibril end, it might not elongate in the presence of PrP monomer unless sonication is done. Also, authors should comment on whether Stx6 fibrils are capable of surface-catalyzed secondary nucleation.

• Although the highlight of the paper is the formation of toxic aggregation intermediates in the presence of Stx6 that prolongs the fibril formation, the report does not describe much about the nature of toxic intermediates. The authors should isolate the intermediate through centrifugation and structurally characterize the intermediate through biophysical experiments (e.g., mass spectrometry, CD, DLS, FTIR, TEM, and Native-PAGE). The authors should perform an MTT assay to check the toxicity of the intermediate apart from the already shown experiment of time-dependent relative neurite length of mouse primary neuron.

[Editors’ note: further revisions were suggested prior to acceptance, as described below.]

Thank you for resubmitting your work entitled "Syntaxin-6 delays prion protein fibril formation and prolongs presence of toxic aggregation intermediates" for further consideration by *eLife*. Your revised article has been evaluated by David Ron (Senior Editor) and a Reviewing Editor.

The manuscript has been improved but there are some remaining issues that need to be addressed, as outlined below:

1) The authors have not addressed the issue of extremely noisy ThT aggregation kinetics correctly, although it is likely that the lag-phase data is robust. However, as the graph is noisy indicating that the aggregation assays are non-standard and do not replicate the behavior of similar assays reported earlier, further explanation from the authors is required.

2) The author's reply to usage of Amylofit fitting software to determine the kinetic parameters is not well justified and the previous concern remains. Authors can regenerate better quality of kinetics data for proper fitting by Amylofit or may explain the limitations of their dataset to such fitting software.

3) It is still unclear whether the physiological prions formed in mouse brains are similar in properties to the ones formed in vitro in NAA conditions by the authors. While the fibrils formed in vivo are quite resistant to PK digestion, can a similar assay be performed on the in vitro formed fibrils? Without any experimental proof, it is difficult to validate if the structure of the fibrils formed in vitro under NAA conditions would be similar to the fibril structures found naturally.

4) The suggested experiment to prove if the intermediate is off-pathway was not performed. The authors are requested to perform this crucial experiment.

---

## [Author Response]

[Editors’ note: The authors appealed the original decision. What follows is the authors’ response to the first round of review.]

Reviewer #1 (Recommendations for the authors):The suggested experiments can be addressed by the authors to make their claim stronger:1) Authors should check in vivo interaction of PrP and Stx6 in different conditions that favors prion pathogenesis. If there is no interaction between Stx6 and Prp, the whole hypothesis of Stx6-mediated enhancement of prion pathogenesis would be baseless.

We agree with the reviewer that the interaction of syntaxin-6 with PrP in the cellular context is a key piece of evidence to support the physiological relevance of our data. We have therefore analysed the interaction between PrP and STX6 in a cell model of prion disease, both in non-infected (PK1) cells and in persistently infected (iF11) cells by FRET imaging. In all cases PrP and STX6 interacted in a perinuclear compartment, most likely the trans Golgi network (TGN). In prion infected cells, we observed additional FRET signals indicating interactions between PrP and STX6 at or near the cell membrane, supporting a role of STX6 in the prion infection process (new Figure 4A, Figures 4S1, 4S2). We also analysed the role of cellular STX6 in PMCA prion replication assays by using brain homogenates of *Stx6* -/- knockout mice (new Figure 4B). Here, the deletion of *Stx6* did not alter the amplification of RML prions, suggesting a role of syntaxin-6 in the initiation of PrP misfolding and/or in the toxicity of misfolded PrP assemblies rather than in the fragmentation / elongation mechanism of prion replication, which corresponds well to our data from experiments in vitro. We have added these points to the discussion of possible mechanisms by which syntaxin-6 alters initial prion formation, prion replication and prion-associated toxicity.

2) Figure 2D: What is the assay end point? Is it 115h? If it is 115h, then figure 2E should also show the status of aggregates at the same time point rather than at 65h.

We have replaced the EM images in Figure 2D and E to better match the endpoint of aggregation of hPrP23 without STX6 (115 h) and to improve contrast in immuno-EM.

3) Figure 5: The reduction in relative neurite length by Prp-Stx6 aggregates compared to PrP fibrils as an indicator of neuro-toxicity although significant but very small in % change. Is this much of change in neurite length with Stx6-PrP aggregates justify the enhanced toxicity model by Stx6? Can authors use some other assays like cell death assays that can show the enhanced toxicity more clearly?

In the new Figure 5S1, we have added neuronal survival measured as % of live neurons compared to t = 0 h as an additional measure of toxicity. Neuronal survival data closely mirror the results from the neurite length assay and are statistically highly significant. It should be noted that the observed 30% drop in neurite length corresponded to a very substantial fraction of neuronal loss. This is likely due to a technical limitation of the automated neurite length assay, in which fragmented neurites of dead neurons are misidentified by the algorithm.

Reviewer #2 (Recommendations for the authors):In this paper, Sangar.et al. discuss an in vitro assay to generate toxic PrP aggregates under near native conditions using purified human and murine PrP (Prpc) which allows them to characterize protein-amyloid interactions. Interestingly, they find that interaction of STX6 (Syntaxin-6) leads to formation of amorphous PrP aggregates instead of fibrillar aggregates which prolongs toxicity in primary neurons. There is novelty in the work but has not been sufficiently explored.Details:1) Does Stx6 bind to preformed prions, or do they bind to the prions only while they are forming? If they do bind to preformed prions their effect on aggregation kinetics may have a different explanation (please see the comment below).

Our immuno-EM and SR-microscopy data indicate that STX6 can both bind to preformed PrP fibrils (Figure 3) and co-aggregate with PrP when present at the outset of the assay (Figure 3S1). When added to preformed fibrils, the protein does not coat the fibril but seems to bind at kinks and fibril ends. We performed a new secondary seeding experiment, using the PrP co-aggregated with STX6 as seeds for a new fibril formation assay (Figure 5S2, now Figure 2S3). The assay revealed that STX6 bound to fibrillar aggregates formed in the plateau phase of aggregation did not significantly impact seeding competence. Similarly, deletion of STX6 from the brain homogenate substrate of PMCA reactions did not alter the amplification of preformed prion seeds (Figure 4B). We therefore conclude that the role of STX6 must lie in the nucleation of fibril formation, as is suggested by the kinetic data the reviewer highlighted. It should be noted that prion replication and PrP fibril formation are not identical processes. Both may be orthogonal in vivo, so that inhibition of PrP amyloid can increase the presence of toxic aggregation intermediates while not affecting prion replication *per se*. We have added this aspect to the discussion of the revised manuscript.

2) Stx6 seems to only increase the lag phase (with a minor effect on the elongation rate) – does it interfere with the nucleation kinetics by binding to the seeds, or do they bind to the folded/unfolded PrP to change the equilibrium and hence alter the nucleation kinetics?Without understanding these basics, it is hard to conclusively state the mechanism of Stx6's effect on PrP aggregation.

Our interpretation of the new seeding data (Figure 5S2, now Figure 2S3) and PMCA prion replication data (Figure 4B) strongly suggest a role in nucleation rather than seed replication through prion mechanisms (fragmentation / elongation) as discussed above. This does not preclude that STX6 may facilitate secondary nucleation and branching as suggested by our imaging data (Figures 2E, 3, 3S1-3), but does imply that these processes do not dominate the kinetics under our assay conditions. This interpretation matches the finding in vivo, where *STX6* is a risk gene specifically for sporadic, not acquired, prion disease.

3) Is stx6 secreted when they are overexpressed? Stx6 overexpression is a risk factor for CJD, so how is the in vivo expression level related to prion formation?

The reviewer raises very relevant questions on the role of STX6 in sporadic Creutzfeldt Jakob disease. Genetic studies have identified two point mutations, which increase STX6 expression, as genetic risk factors for sCJD. Specifically, *STX6* mRNA levels are higher in cells implicated in prion replication in vivo. Our collaborator Simon Mead has found that Stx6 knockout moderately delays onset of prion disease in RML-infected mice (Jones et al. bioRxiv, https://doi.org/10.1101/2023.01.10.523281) and will explore questions of STX6 expression and trafficking in vivo in forthcoming manuscripts.

4) How do the authors connect the expression level correlation to PrP toxicity to formation of PrP/Stx6 mixed aggregates?

As we discuss in the manuscript and in our response to the reviewer’s overall assessment, STX6 delays the formation of mature amyloid fibrils, which coincides with a prolonged presence of neurotoxic PrP aggregates. The nature of the neurotoxic species in prion disease – and for that matter in Alzheimer’s and other amyloid diseases – is the subject of intense research in our institute and elsewhere. We cannot claim to know its exact nature and we did not mean to imply that the co-aggregates of PrP and STX6 are necessarily toxic themselves. Our argument follows the well-established toxic oligomer hypothesis (see: Haass & Selkoe 2007), so our data imply that by delaying formation of mature amyloid fibrils, STX6 causes toxic aggregation intermediates to persist. We have edited our discussion to make this point more clearly.

5) Is there any evidence that PrP/Stx6 forms mixed amyloids in vivo?

Our new FRET interaction data from prion-infected cell models (Figures 4A, 4S1, 4S2) imply that both proteins interact in the trans Golgi network and, specifically in prion infected cells, in a membrane associated compartment, which does suggest interaction of STX6 with misfolded PrP species in these infected cell models. In their forthcoming paper, our collaborator Prof. Simon Mead presents immuno-histological data from scrapie infected mice to further assess STX6 deposition in vivo (Jones et al. bioRxiv, https://doi.org/10.1101/2023.01.10.523281).

6) How do the authors deconvolute the toxicity of PrP and Stx6 in the PrP/Stx6 mixed amyloids? The toxicity could be due to Stx6 or PrP aggregates/amyloids?

As we discussed in response to point 4 above, the nature of the toxic species in prion and other neurodegenerative protein misfolding diseases is yet unknown. However, from our in vitro experiments (Figure 5 and 5S1) we can exclude the hypothesis that STX6 itself is neurotoxic under our assay conditions.

7) The toxicity differences are very modest given stark differences in aggregate morphology in the presence or absence of Stx6. How do the authors explain this?

Please refer to our response to reviewer 1, point 3. In brief, the neurite length assay underreports the differences in toxicity. We added neuronal survival data (Figure 5S1) from the same experiment, which shows that PrP species from early aggregation time points are highly toxic and kill up to 80% of neurons. Neurotoxicity decreases with time as amyloid fibrils form, but persists longer in the presence of STX6 than in its absence.

Other comments:1) Apart from the development of an in vitro setup to generate toxic PrP aggregates, a major finding of the manuscript is the interaction between STX6 and PrP which prevents formation of fibrillar aggregates supported by ThT fluorescence assay, immunogold-EM analysis, and high-resolution microscopy and cytotoxicity data using primary neurons. From the data shown here we can convincingly conclude the following:a) Recombinant PrP forms fibrillar aggregates in the assay conditions described.b) STX6 and PrP interact in vitro.c) This interaction prevents formation of fibrillar aggregates.

We agree with the reviewer’s interpretation, but would amend that STX6 delays rather than completely prevents fibril formation at sub-stoichiometric ratios.

2) However, the data to show that STX6 interacts with PrP in-vivo (both in its monomeric and aggregated form) is lacking. The co-localization of Syntaxin-6 in the PrP aggregates isolated from clinical samples (patient samples) would be a crucial validation of the observations made in-vitro i.e, Interaction of Syntaxin-6 with PrP aggregates. While this may be complicated, it will be difficult to interpret the observations made in vitro in the absence of in vivo data.

We agree with the reviewer that in vivo interaction data are key to demonstrating the physiological relevance of STX6 in prion disease. As discussed above, we have therefore performed FRET imaging in scrapie-infected murine cells to demonstrate interaction between both proteins in vivo. The forthcoming corresponding study from the Mead laboratory (https://doi.org/10.1101/2023.01.10.523281) shows histopathological data of STX6 deposition in scrapie infected mice. A histopathological analysis of brain autopsies of CJD patients lies beyond the scope of the biophysical study presented in this manuscript.

3) And more importantly, the article could be further strengthened by more robust biochemical analysis into the mechanism of how STX6 interacts with PrP.

We fully agree with the reviewer on this point. The native aggregaton assay developed in this manuscript now provides us with the tools to do so, and a NMR-based study on the molecular interaction between STX6 and PrP will be the subject of a forthcoming manuscript.

4) Following up on point 3, the authors may investigate whether STX6 interaction facilitates release of PrP from cells similar to tau (Lee et al. 2021).

This is an excellent suggestion. The cell to cell transfer of prions is still poorly understood and is an active topic of ongoing research in the MRC Prion Unit.

5) In the TEM micrographs, it is very hard to locate the aggregates (Figure 2F and S4B). But for the ease of understanding of general reader, it is important to show presence of both the aggregate and the colocalization of STX6 in the immunogold staining.

We have replaced EM micrographs in Figures 2, 3 and S4 (now: Figure 2S1) to improve contrast, which hopefully resolves the issue raised by the reviewer. Interaction between both proteins can also be clearly seen in super-resolution fluorescence microscopy images (Figures 3 and 3S1) and in the FRET data (Figures 4, 4S1, 4S2).

6) Does treatment of primary neurons with 1:1 stoichiometry of STX6 and PrP prevent PrP induced neurotoxicity? Concentration dependent analysis may be done. This will further validate the finding observed in Figure S2(A).

Unfortunately, it was not possible to perform the toxicity assay on primary neurons at 1:1 stoichiometry under our assay conditions, since STX6 itself became toxic at these very high concentrations. We did, however, add data on the number of live neurons as an additional marker of neurotoxicity to validate our results (Figure 5S1). We found that neuronal survival data fully correlated to neurite length data and demonstrated a high degree of neuronal loss (~80%) when incubated with early PrP aggregation intermediates.

7) While this may be out of scope of the study, authors may look into the effects of overexpression or knockdown of syntaxin-6 on PrP aggregate formation.

We have included data from in vitro prion amplification assays (PMCA) in the revised manuscript, in which we compare amplification of RML mouse prions in a substrate of WT mouse brain homogenate with that of *Stx6* -/- mice. Interestingly, *Stx6* knockout did not alter the amplification of pre-existing prion seed (Figure 4B) under these conditions. This hypothesis is supported by the new experimental data of a secondary seeding assay, in which we assessed the seeding competence of PrP assemblies at different incubation times in the presence and absence of STX6 (Figure 5S2, now Figure 2S3). The incubation times and seed samples were identical to those used in neurotoxicity assays. We found that, analogous to its effect on toxicity and ThT fluorescence, syntaxin-6 delayed, but did not prevent the formation of seeding competent PrP assemblies. These data support our interpretation that syntaxin-6 likely alters primary nucleation and prolongs the presence of early, non-fibrillar aggregation intermediates, while it has little influence on the fibril fragmentation/elongation step of prion replication.

Reviewer #3 (Recommendations for the authors):The objectives of the project are well-addressed and the discussion part is nicely illustrated. Some of the results look very interesting, leading to some exciting predictions (e.g., refolding of monomer, formation of off-pathway intermediates, etc.) based on the initial result. However, some additional experiments are required for a better understanding of the present results. Hereby, the authors are requested to address the following concerns and perform additional experiments that are mentioned in the following section.• Although the fibrillation assay used in this manuscript does not use denaturant or detergent, however, the PrP proteins are kept at 42{degree sign}C, pH 6.8, under agitation with zirconium 86 beads. How a condition like agitation can be claimed as a native environmental condition and will ensure that the assay condition is not facilitating protein unfolding, unlike the other reported fibrillation assays?

The reviewer raises a valid point that there is no direct physiological correlate to agitation by stirring beads in vivo. There have been arguments that parts of the chaperone system, i.e. Hsp104 in yeast, which can fragment amyloid fibrils, have a similar function in vivo to the physical fragmentation through stirring beads. Systematic studies involving agitation (eg. Cohen et al. PNAS 2013; https://doi.org/10.1073/pnas.1218402110) have concluded that increasing agitation shifts secondary replication mechanisms to a prion-like fragmentation-elongation mechanism, which is compatible with our observations and with the results from PMCA prion amplification assays, which we have now included in Figure 4B. While it is difficult to deconvolute aggregation from protein misfolding, we see no evidence for PrP denaturation as response to agitation under our assay conditions. We furthermore find that, while PrP didn’t form fibrils in the course of a typical experiment when agitated without stirring beads, fibril formation did not depend on the specific surface of the bead, as glass beads were as effective, or more, as Zr beads in facilitating fibril formation (new Figure S2C).

• In line 121, the authors wrote that "formation of aggregates with more amorphous morphology, which seems to consist of short fibrils" this is inconsistent comment as proteins will form amorphous or amyloids. Amorphous protein aggregates can't compose of short fibrils?

We apologize for the misleading wording. We were trying to express that PrP formed aggregate clusters rather than single amyloid fibrils. These clusters retained a fibril-like substructure and exhibited ThT fluorescence. We have clarified this in the text. (see: Figures 2D and E, Figures 3, 3S2, 3S3).

• The Thioflavin T aggregation profile looks extremely noisy (e.g., Figure 1A, 2A) with huge fluctuation in the Y axis. The authors should explain the reason for this much fluctuation in the ThT signal and fit the kinetic profile taking care of the standard deviation.

The noise of the aggregation curves is most likely due to the formation of very large aggregates and the presence of the stirring beads. However, our kinetic data all resulted from the individual fitting of replicate wells. While the fluctuations may add some uncertainty to the ThT fluorescence amplitude, they had very little effect on the precision of lag phase determination, which was our primary readout.

• The authors have calculated lag times (t50) and fibril elongation rate (k) in different experimental conditions throughout the manuscript. However, the authors did not show the calculation performed to determine these two parameters in the method section. The authors should incorporate that. Moreover, the authors should perform a web-based global fitting software interface, such as Amylofit (Meisl et al., 2016), to precisely evaluate different kinetic parameters for better comparison.

We apologize for the omission. Parameters were determined by a simple sigmoidal fit; we have added the fitted equation to the method section. We also attempted to analyse the kinetics through the AmyloFit global fitting model and discussed our results with Dr Georg Meisl, one of the lead developers of AmyloFit. Unfortunately, at the moment the AmyloFit framework is not suited to the analysis of folded proteins, which have a pre-equilibrium of a native and an aggregation competent state, so that we could not use it to further analyse or data.

• The quality of TEM images of hPrP (Figure 1C) and moPrP (Figure S2C) is of poor quality. The authors are requested to change these with high-resolution images. Moreover, the present immune-gold staining image (Figure 2F) does not show the fibril where immune-gold labeled Stx6 is incorporated. The authors should provide a better-quality immune-gold staining image with a clear fibril image so that the location of incorporated Stx6 (immune-gold labeled) is clearly understood.

We apologize for the suboptimal quality of immuno-EM images. We have replaced micrographs in Figures 2, 3 and 2S1 with higher contrast images.

• The authors have shown that full-length PrP proteins form amyloid fibrils under near-native conditions. The authors should centrifuge to isolate the fibril fraction and show the proteinase-K digestion of the fibril, which is a known property of PrP amyloid fibril. The authors also should compare this PK resistance data with native PrP amyloid isolated from mouse prion strains ME7 and RML that the authors have used. Further, the authors should also perform the same with the fibril bound to Stx6, Hspa1a and Stathmin 1 to understand the nature of the aggregates in the presence of the additives.

We performed several PK digestion assays on fibrils formed in NAA when seeded either with recombinant fibrils from a previous round of NAA or with prion rods isolated from RML or ME7 scrapie infected mice. PrP fibrils formed via NAA showed a substantial (~50%) resistance to PK digestion in ThT assays and a PK-resistant PrP fragment band when digested with PK at 5 µg/mL concentration. (see Author response image 1).

Interestingly, we found that PrP fibrils formed under NAA conditions proved to be highly resistant to SDS denaturation. Unfortunately, this prevented us from characterizing their PK resistance via SDS-PAGE in a more thorough fashion, since we could not be sure that PrP bands in SDS-PAGE were representative of the NAA fibril population as a whole. We are therefore presenting these data here in Author response image 1 for the reviewer’s information rather than including them in the revised manuscript. The structural characterization of NAA fibrils when compared to authentic prion rods will be the subject of a forthcoming study.

**Author response image 1. sa2fig1:** Murine (mPrP23) fibrils formed in NAA show limited proteinase K resistance. (A) Baseline-corrected ThT fluorescence of endpoint NAA fibrils digested by PK (5 µg / mL). NAA fibrils had been formed in NAA assays seeded with recombinant fibrils (NAA), RML or ME7 prion rods. A ThT fluorescence baseline was recorded for 1.5 h, PK was added and AEBSF protease inhibitor was then added after 5 h incubation. Curves represent means ± SD of triplicate experiments. (B) Silver stained SDS-PAGE of the samples from panel A.

• The authors have speculated the formation of the "off-pathway" intermediate based on the observation that the lag phase of aggregation has increased and ThT amplitude has dropped at high monomer concentration. The author should isolate the intermediate through centrifugation at the particular aggregation condition where the predicted off-pathway intermediate populates the most. To precisely comment on the "off-pathway" intermediate, the authors should add the isolated intermediate fraction during the fibrillation kinetics in a dose-dependent manner to see the concentration effect of the intermediate in fibrillation kinetics that will provide direct evidence to support the statement.

The reviewer raises a very valid point, which we addressed in two new experiments: (1) Figure 2S2 analyzes the solubility of rPrP23 at different time points of the native aggregation assay corresponding to the time points at which neurotoxicity was assessed. We found that the presence of soluble PrP intermediates coincides with the time points of maximal toxicity. Syntaxin-6 prolongs the presence of toxic PrP species as it prolongs the presence of soluble intermediates, but, unlike Hsp70, it did not prevent PrP aggregation. (2) We assessed the seeding competence of PrP aggregation intermediates in a secondary seeding assay (Figure 5S2, now Figure 2S3). Here, syntaxin-6 delayed the formation of seeding competent assemblies. The formation of seeding competent assemblies coincided with a decrease in toxicity. Taken together, these data support our interpretation that STX6 delays the formation of fibrillar seeding competent assemblies and prolongs the presence of pre-fibrillar toxic PrP species.

• To check the interaction with PrP protein in super-resolution microscopy, the authors have chosen only Stx6 among HSPA1A, STMN1, and STX6. The authors should justify the specific reasons behind choosing Stx6 over the others.

We chose STX6 as our main protein of interest and as a test case for our newly developed native aggregation assay because it was recently identified as a risk factor for sporadic CJD. We included Hsp70 (HSPA1A) as a known inhibitor of amyloid fibril formation. STMN1 was included as a negative control. It has a highly α-helical structure, similar to STX6 and was previously identified in our lab as a potential inhibitor of α-synuclein fibril formation. This rationale has now been stated explicitly in the text.

• The authors have observed that the interaction sites of Stx6 were often located at fibril ends or kinks in the fibril. Thus, the authors have proposed a potential role of Stx6 in fibril breakage or secondary nucleation. Firstly, the authors should reconfirm the Stx6 binding location in PrP from the clear immune-gold staining images. Secondly, the authors should isolate the Stx6-bound fibrils and perform a seeding reaction (prion amplification) with only the PrP monomer to see whether Stx6-bound PrP fibrils are competent for elongation or not. If Stx6 binds to the fibril end, it might not elongate in the presence of PrP monomer unless sonication is done. Also, authors should comment on whether Stx6 fibrils are capable of surface-catalyzed secondary nucleation.

We thank the reviewer for this suggestion. We have performed both experiments, as detailed above (Figures 2S2 and 5S2) and included higher contrast immuno-EM images (Figure 3F). In addition we performed PMCA amplification of infectious RML prion seeds in brain homogenate substrates from WT mice and from *Stx6 -/-* knockout mice (Figure 4B). In all cases, STX6 had no influence on the seeding competence of fibrillar aggregates. So, while STX6 seems to preferentially bind to the ends of fibrils, it does not seem to prevent elongation, but rather affects early aggregate formation of PrP. Our imaging data suggest that binding of STX6 may also facilitate branching and secondary nucleation. However, prion-like replication (fragmentation / elongation) dominates under PMCA conditions, so that deletion of cellular *Stx6* had no effect on the replication efficiency. We have included these aspects in our revised discussion to narrow down the mechanistic implications of our results.

• Although the highlight of the paper is the formation of toxic aggregation intermediates in the presence of Stx6 that prolongs the fibril formation, the report does not describe much about the nature of toxic intermediates. The authors should isolate the intermediate through centrifugation and structurally characterize the intermediate through biophysical experiments (e.g., mass spectrometry, CD, DLS, FTIR, TEM, and Native-PAGE). The authors should perform an MTT assay to check the toxicity of the intermediate apart from the already shown experiment of time-dependent relative neurite length of mouse primary neuron.

We have assessed the relative numbers of live neurons as an orthogonal toxicity assay and added these data in the new Figure 5S1. Toxicity data from neuronal survival closely mirrored those from the neurite length assay.

While we appreciate that the lack of solid structural and biophysical data on the nature of the toxic species in prion disease (and other protein misfolding diseases) is frustrating, characterizing the nature of the toxic species is well beyond the scope of the current manuscript. The reviewer is doubtlessly aware that it is one of the key unsolved questions in all amyloid disease and therefore one of the top research priorities for the MRC Prion Unit, as well as many other institutes around the world. If we are successful in our characterization, we will certainly publish these results in a prominent journal in a timely manner.

[Editors’ note: what follows is the authors’ response to the second round of review.]

The manuscript has been improved but there are some remaining issues that need to be addressed, as outlined below:1) The authors have not addressed the issue of extremely noisy ThT aggregation kinetics correctly, although it is likely that the lag-phase data is robust. However, as the graph is noisy indicating that the aggregation assays are non-standard and do not replicate the behavior of similar assays reported earlier, further explanation from the authors is required.

Thank you for highlighting this issue. Fluctuations appeared to be caused by the formation of large aggregate clusters and were more prominent at high PrP concentrations. These experiments also saw a faster drop-off in fluorescence signal after the plateau phase was reached (see: Figure 2 —figure supplement 2 vs. Figure 2- FS 3) We did explore strategies to suppress the fluctuations in ThT fluorescence traces in the additional secondary seeding experiments performed in response to comment 4 (Figure 2 – FS 4 and 5). Here, fluctuations could be minimized by optimizing the height of fluorescence detection in the plate. To address the concern whether fluorescence fluctuations would change basic kinetic parameters such as lag time, we performed a moving average filtering of the kinetic data in the concentration dependent aggregation (Figure 1E) before analysis in Amylofit (Figure 1 – FS 4). As the reviewer correctly predicted, the noise had no effect on the analysis of lag times when comparing the filtered (Figure 1 – FS 4B) and unfiltered data (Figure 1F), so that we decided to keep the original kinetic data rather than repeating experiments for a large part of the study. We now discuss fluorescence fluctuations in ll. 119-126 of the revised manuscript.

2) The author's reply to usage of Amylofit fitting software to determine the kinetic parameters is not well justified and the previous concern remains. Authors can regenerate better quality of kinetics data for proper fitting by Amylofit or may explain the limitations of their dataset to such fitting software.

At the reviewer’s suggestion, we performed a global fitting analysis of concentration-dependent aggregation in Amylofit (new Figure 1 —figure supplement 4). Unlike previous versions of the software, Amylofit can now incorporate a saturated elongation process into the fitting model, which made it feasible to analyse our data in that framework. As we had predicted, all kinetic models that did not include saturated elongation completely failed to fit our experimental data. In contrast a simple fragmentation model with saturated elongation fitted our data adequately (Figure 1 – FS 4C-F). We did test more elaborate models that included saturated elongation, but they did not improve the fit and were therefore not included in the manuscript. In conclusion, Amylofit supported our conclusion that hPrP23 aggregation kinetics under native conditions are rate-limited by a 0^th^ order process with regard to PrP concentration, most likely the structual rearrangement from PrP^C^ to the amyloid conformer.

3) It is still unclear whether the physiological prions formed in mouse brains are similar in properties to the ones formed in vitro in NAA conditions by the authors. While the fibrils formed in vivo are quite resistant to PK digestion, can a similar assay be performed on the in vitro formed fibrils? Without any experimental proof, it is difficult to validate if the structure of the fibrils formed in vitro under NAA conditions would be similar to the fibril structures found naturally.

We thank the reviewer for the suggestion and we have now added a proteinase K digestion experiment under conditions used for PrP^Sc^ detection (50 µg/ml, 30 min, 37°C; Figure 1D, Figure 1 —figure supplement 2H). We found NAA fibrils to be as resistant to PK digestion as authentic RML prion rods isolated from scrapie-infected mice. However, PK digestion resulted in a fragment of ~ 12 kD, which is considerably smaller than the PK-resistant fragment of PrP^Sc^. Whether the fibril structure of NAA products is identical to a part of the prion rod structure will be determined in a future cryo-EM study.

4) The suggested experiment to prove if the intermediate is off-pathway was not performed. The authors are requested to perform this crucial experiment.To precisely comment on the "off-pathway" intermediate, the authors should add the isolated intermediate fraction during the fibrillation kinetics in a dose-dependent manner to see the concentration effect of the intermediate in fibrillation kinetics that will provide direct evidence to support the statement.

We have added two secondary seeding assays to probe the nature of the aggregates formed in the presence of syntaxin-6. The second assay is the one suggested by the reviewer, from which we can conclude that any PrP aggregates formed in the presence of syntaxin-6 prior to the formation of ThT-positive fibrils are reduced in seeding capacity by at least three orders of magnitude when compared to untreated PrP and that soluble aggregation intermediates are not seeding competent (ll. 112 -126):

“To probe the nature of PrP aggregates formed in the presence of syntaxin-6, we performed two secondary seeding assays. In the first assay, we tested the seeding capacity of PrP samples harvested at different incubation times (20, 40, 60, 90 h) in the presence and absence of syntaxin-6 in a NAA seeded with 0.1% fibrillar hPrP23 (Figure 2 —figure supplement 3). Samples harvested during the lag phase of either reactions were not seeding competent, but both aggregates formed in the presence and absence of syntaxin-6 (1:10) were seeding competent after the formation of ThT-positive aggregates, indicating that, at sub-stoichiometric ratios, syntaxin-6 delays, but does not prevent the formation of seeding competent PrP fibrils. We then generated seeds in a second experiment, in which the primary seed was added at a lower concentration (0.01%), which prolonged the delay in fibril formation by syntaxin-6 (Figure 2 —figure supplement 4A). Seed preparations harvested after 70 h incubation were then separated into total, soluble and insoluble fractions and added to secondary seeding assays at 10^-3^ to 10^-8^ molar ratio monomer equivalents (Figure 2 —figure supplement 4B-E). Both in syntaxin-6 and in untreated samples seeding activity was only found in the total and insoluble fractions. The presence of syntaxin-6 lowered the amount of seeding competent aggregates by at least three orders of magnitude (Figure 2 —figure supplement 4C-E and 5).”